*Report*

EMBO
*reports*

# YY1 binding is a gene-intrinsic barrier to Xist-mediated gene silencing

Joseph S Bowness [1,2], Mafalda Almeida [1], Tatyana B Nesterova [1] & Neil Brockdorff [1✉]

## Abstract

**X chromosome inactivation (XCI) in mammals is mediated by Xist RNA which functions in *cis* to silence genes on a single X chromosome in XX female cells, thereby equalising levels of X-linked gene expression relative to XY males. XCI progresses over a period of several days, with some X-linked genes silencing faster than others. The chromosomal location of a gene is an important determinant of silencing rate, but uncharacterised gene-intrinsic features also mediate resistance or susceptibility to silencing. In this study, we examine mouse embryonic stem cell lines with an inducible *Xist* allele (iXist-ChrX mESCs) and integrate allele-specific data of gene silencing and decreasing inactive X (Xi) chromatin accessibility over time courses of Xist induction with cellular differentiation. Our analysis reveals that motifs bound by the transcription factor YY1 are associated with persistently accessible regulatory elements, including many promoters and enhancers of slow-silencing genes. We further show that YY1 is evicted relatively slowly from target sites on Xi, and that silencing of X-linked genes is increased upon YY1 degradation. Together our results suggest that YY1 acts as a barrier to Xist-mediated silencing until the late stages of the XCI process.**

**Keywords** X Inactivation; Transcription Factor; Gene Silencing; Chromatin; Xist
**Subject Category** Chromatin, Transcription & Genomics

## Introduction

In mammals X chromosome inactivation (XCI) ensures genes on one X chromosome are silenced to compensate X-linked gene dosage between XX and XY cells. XCI is initiated during early embryonic development by the expression of Xist, a long non-coding RNA, from the future inactive X (Xi) (Brockdorff et al, 1992; Brown et al, 1992; Kay et al, 1993). Xist RNA spreads over the chromosome in *cis* and recruits molecular pathways which collectively modify the underlying chromatin from an active to a repressive state, resulting in complete transcriptional silencing of most X-linked genes (for a recent review, see Loda et al, 2022). An

important subset of X-linked genes, including those present in the pseudoautosomal regions shared with the Y chromosome, escape inactivation to varying degrees (Lyon, 1962; Carrel and Willard, 2005; Yang et al, 2010). Although there are differences in the developmental regulation and timing of XCI between mammalian species (Dupont and Gribnau, 2013), many molecular mechanisms of Xist-mediated silencing are conserved, and insights derived from studying tractable experimental models such as mouse embryonic stem cell (mESC) lines often extrapolate to human XCI and have relevance to diseases linked to the X chromosome.

Previous studies have established that the kinetics of Xist-mediated gene silencing vary considerably between X-linked genes (in the range of several hours to days) and report comparable categories of fast and slow-silencing genes despite using a variety of in vivo and in vitro model systems and different assays to quantify silencing (Lin et al, 2007; Marks et al, 2015; Borensztein et al, 2017; Loda et al, 2017; Barros de Andrade E Sousa et al, 2019; Pacini et al, 2021; Bowness et al, 2022). Analysis of the features conferring different silencing rates have revealed correlations with proximity to the *Xist* locus (Lin et al, 2007; Kelsey et al, 2015; Barros de Andrade E Sousa et al, 2019; Nesterova et al, 2019), and to chromosomal sites where Xist RNA preferentially localises (Engreitz et al, 2013; Markaki et al, 2021). In addition, slow-silencing genes are more likely to have a high expression level on the active X (Xa) chromosome, i.e., prior to the onset of Xist-mediated silencing (Loda et al, 2017; Nesterova et al, 2019). This observation highlights that gene-intrinsic features have a contributory role in determining rates of gene silencing in XCI. Relatedly, escape from XCI is also mediated by gene-intrinsic factors (Peeters et al, 2018, 2023; Fang et al, 2019a) in a non-trivial manner as catalogues of escapee genes vary substantially between different tissues (Berletch et al, 2015; Andergassen et al, 2017). Studies focusing on escape have implicated roles for CTCF binding (Filippova et al, 2005), local chromatin status (Calabrese et al, 2012), or proximity to tissue-specific enhancer elements (Andergassen et al, 2017). However, a full understanding of gene-intrinsic features affecting both silencing kinetics and escape is lacking.

In this study, we set out to investigate if promoter/enhancer binding by specific transcription factors (TFs) contributes to gene-intrinsic variation in silencing rates. Initially, we focused on chromatin accessibility because, in addition to denoting the locations of *cis*-regulatory elements (REs) in the genome (Thurman et al, 2012), it conveys information about the relative activity levels of REs as promoters, enhancers or insulators, and marks instances

[1]Department of Biochemistry, University of Oxford, Oxford OX1 3QU, UK. [2]Present address: Centre for Genomic Regulation (CRG), The Barcelona Institute of Science and Technology, Dr. Aiguader 88, 08003 Barcelona, Spain. ✉E-mail: neil.brockdorff@bioch.ox.ac.uk

where TFs bind their cognate motif sequences in REs (see reviews (Klemm et al, 2019; Kim and Wysocka, 2023)). Accordingly, we profiled the rate of chromatin accessibility reduction from Xi at X-linked REs using ATAC-seq (Buenrostro et al, 2013), and used this to investigate TF binding sites that correlate with differential rates of silencing. Our analysis revealed a strong link between slow silencing kinetics and the presence of binding sites for the TF Yin Yang 1 (YY1). Consistent with this observation we found that YY1 is evicted particularly slowly from Xi in an extended time course of Xist RNA induction, and moreover that depletion of YY1 leads to an increased level of Xi silencing at multiple timepoints.

## Results and discussion

To assess the contribution of TF binding in determining the rate of Xi gene silencing we made use of an established interspecific XX mESC model, iXist-ChrX, in which Xist RNA expression from a single X chromosome is driven by the addition of doxycycline (Nesterova et al, 2019). In recent work we used this system to characterise Xi gene silencing over a time course of mESC to neuronal precursor cell (NPC) differentiation, in the process defining classes of fast, intermediate and slow-silencing genes on the basis of their rates of silencing (Bowness et al, 2022). For the present study we extended this characterisation by performing equivalent time courses of ATAC-seq in iXist-ChrX$_{Dom}$ cells, which carry inducible Xist on the *Mus musculus domesticus* allele and a recombination event which makes only the 103 Mb proximal to *Xist* informative for allelic analysis (Nesterova et al, 2019). In Fig. 1, we compare allele-specific ATAC-seq data with chromatin-associated RNA sequencing (ChrRNA-seq) datasets of gene silencing previously collected from this line under the same experimental conditions. As shown in representative genome tracks in Fig. 1A, most REs on Xi show a progressive reduction in chromatin accessibility over the time course of XCI. This can be quantified for individual elements at each timepoint by taking reads overlapping strain-specific SNPs and calculating the fraction of allelic reads mapping to the chromosome with inducible Xist ($Xi / (Xi + Xa)$). When "allelic ratios" are calculated for all ATAC-seq peaks amenable to allelic analysis ($n = 790$ of 2042 ChrX peaks, see "Methods"), there is a clear trend of decrease from biallelic accessibility of REs in uninduced mESCs (median ~0.50) to monoallelic accessibility in differentiated NPCs (median ~0.11) (Fig. EV1A). A few exceptions demonstrate increasing accessibility on Xi (Fig. EV1B), for example Xist and the clusters of CTCF-binding REs at the *Firre* and *Dxz4* loci implicated in the 3D superstructure of the Xi chromosome (Rao et al, 2014; Deng et al, 2015; Yang et al, 2015; Giorgetti et al, 2016).

On average, allelic ratios of RE accessibility decrease at a comparatively slower rate than gene silencing quantified by the same metric from ChrRNA-seq data (Fig. EV1C). This slower rate is unlikely to be fully accounted for by subsets of REs remaining persistently accessible throughout XCI, since for individual genes the allelic ratio of accessibility at its promoter decreases more slowly than it silences (Fig. 1B). It is important to note that we do not necessarily interpret slower kinetics to mean that loss of promoter accessibility is mechanistically downstream of gene silencing. Chromatin accessibility and gene expression are not

correlated linearly in other dynamic developmental processes (Starks et al, 2019; Kiani et al, 2022; Tu et al, 2023) and both may be influenced by Xist RNA via a variety of indirect or direct mechanisms (Pintacuda et al, 2017; Jégu et al, 2019).

Using the same methodology as we previously applied to gene silencing (Bowness et al, 2022), we fit REs on the X chromosome with exponential decay curves tracing the trajectory of decreasing Xi accessibility over the time course of Xist induction. This enabled quantification of accessibility reduction kinetics by a halftime ($t_{1/2}$) value of the inferred time taken for Xi accessibility to decrease to half its initial value. Halftimes were calculated for the majority of REs (657/790), although not applicable for elements whose accessibility did not decrease by half over the time course. A general chromosome-wide comparison of halftimes showed that on average distal (non-promoter) REs, with the exception of those bound by CTCF, lose accessibility faster than promoters (Fig. EV1D).

When comparing between genes, we noticed a clear correspondence between the rate of accessibility decrease and gene silencing, such that genes which fall into categories of fast, intermediate or slow- silencing broadly retain this same order when compared by the allelic ratio decrease of their promoter accessibility (Fig. 1B). Consistent with this, accessibility reduction halftimes are correlated with gene silencing halftimes both for promoters ($R = 0.64$, Spearman correlation) and distal elements simplistically assigned to their "nearest" gene by linear genomic proximity ($R = 0.46$) (Fig. 1C). The correspondence is also clear when REs are grouped by the silencing category (i.e.. fast, intermediate or slow) of their nearest gene, further demonstrating that REs associated with slow-silencing genes tend to lose accessibility significantly more slowly than those in proximity to fast-silencing genes (Fig. 1D).

Given that chromatin accessibility is a signature of TF binding, we hypothesised that specific transcription factor(s) may be persistently bound at regulatory elements which have slow accessibility reduction kinetics during XCI. By ranking REs according to accessibility reduction halftimes, we made equal-sizes groups of "depleted" ($t_{1/2} < 6$ days) and "persistent" ($t_{1/2} > 6$ days or allelic ratio > 0.25 in NPCs) REs (Fig. 2A) and performed motif enrichment analysis to identify TF motifs enriched in one group compared to the other (Dataset EV1). The most significant motif is YY1, which is enriched approximately 3-fold in persistent peaks compared to depleted peaks (Fig. 2B). A second motif enriched within the persistent group is CTCF, an expected result given that most of the aforementioned architectural CTCF sites within *Dxz4* and *Firre* clusters are classified as persistent REs in this analysis (Fig. EV1B).

We validated the association between YY1 binding and persistent chromatin accessibility by overlapping X-linked REs with bona fide peaks of YY1 binding identified by ChIP-seq in iXist-ChrX mESCs using a commercial monoclonal antibody against endogenous YY1 protein (Fig. 2C). YY1-bound REs are overrepresented within the "persistent" group (Fig. 2D) and have significantly slower accessibility reduction halftimes compared to REs which do not overlap with YY1 peaks (Fig. 2E). Extending this analysis, we examined if YY1 binding is also associated with slow gene silencing. Indeed, X-linked genes defined as "direct YY1 targets" by the presence of a YY1 peak directly overlapping their TSS are strongly associated with slow silencing both in terms of gene categories (Fig. 2F) and halftimes (Fig. 2G). This is unlikely to

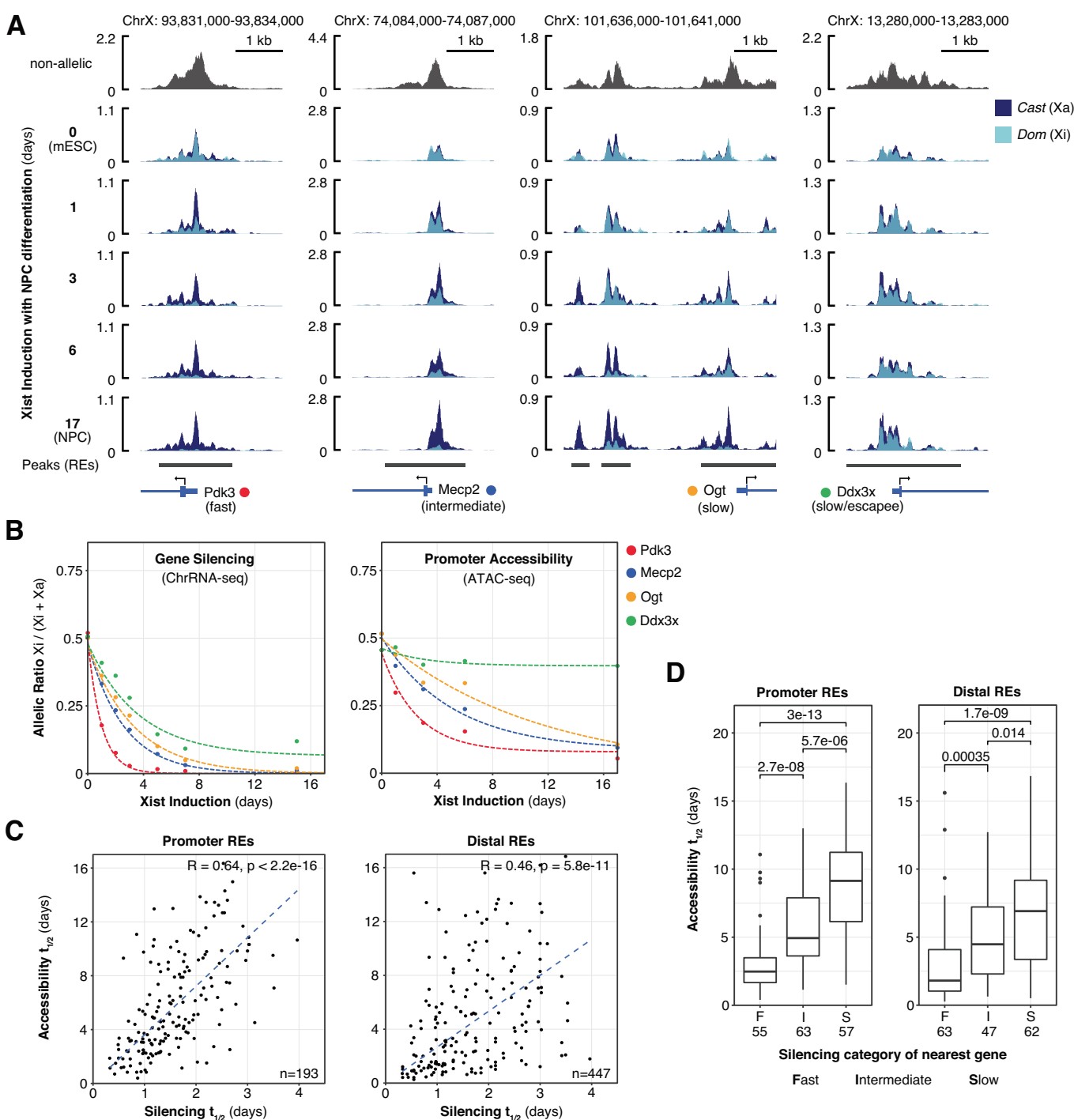

**Figure 1. Corresponding dynamics of gene silencing and decreasing Xi chromatin accessibility.**

(A) ATAC-seq genome tracks at promoter regions of four X-linked genes showing allele-specific changes to Xi accessibility over a time course of Xist induction with NPC differentiation. Replicates averaged using SparK software (Kurtenbach and Harbour, 2019) ($n = 3$ for 0 h mESC, $n = 2$ per NPC timepoint). (B) Trajectories of gene silencing (left) and accessibility of the promoter regions (right) for the four X-linked genes displayed in (A). Lines trace exponential decay curves fitted to each individual gene or regulatory element (RE). Allelic ratios are averages of $n = 3$ or $n = 2$ replicates per timepoint (ChrRNA-seq data from GSE185843). (C) Scatter plots comparing halftimes of RE accessibility and silencing of their putative target genes. $R$ values represent Spearman's rank correlation coefficients. (D) Boxplots of RE accessibility halftimes grouped by the silencing category of their putative target genes. Boxes span first to third quartiles with a central line indicating the median. Whiskers extend to 1.5 * the interquartile range and outliers outside this range are plotted as separate points. Significance ($P$ values) calculated by Wilcoxon rank-sum test. Numbers of REs in each category are indicated below.

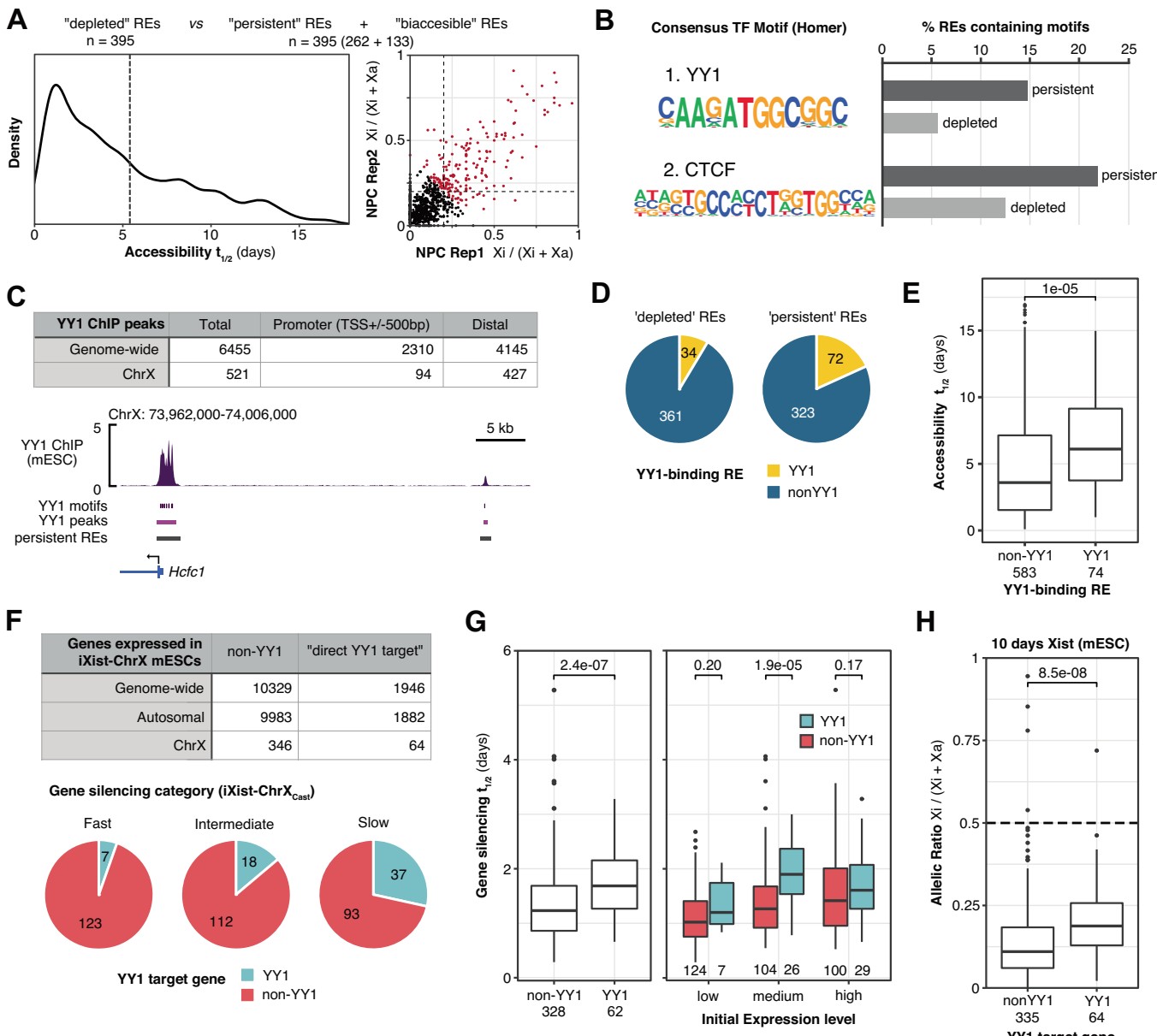

be solely attributable to a correlation between YY1 binding and high levels of initial gene expression, as slower silencing of direct YY1 targets is particularly pronounced within the group of genes defined as "medium-expressed" (Fig. 2G). In addition, we previously reported that undifferentiated iXist-ChrX mESCs subjected to prolonged Xist induction fail to establish complete gene silencing in the absence of cellular differentiation (see Fig. 4 in Bowness et al, 2022). Reanalysis of this data revealed that YY1 target genes show significantly higher residual expression from Xi at this "plateau" of silencing in undifferentiated mESCs (Fig. 2H), corroborating an association between YY1 binding and gene-intrinsic resistance to Xist-mediated silencing.

Next, we directly investigated if YY1 remains bound to its target sites on Xi late into the XCI process, potentially in opposition to pathways of Xist-mediated silencing. We engineered iXist-ChrX

cells using CRISPR-Cas9-mediated homologous recombination to knock-in a FKBP12$^{F36V}$-2xStrep-3xT7 fusion tag (Nabet et al, 2018; Dimitrova et al, 2022) to the C-terminal of endogenous YY1 in iXist-ChrX cells. We generated YY1-FKBP12$^{F36V}$-2xStrep-3xT7 lines (hereafter YY1-FKBP12$^{F36V}$) in both iXist-ChrX$_{Cast}$ (Fig. 3A) and iXist-ChrX$_{Dom}$ (Fig. EV2A) backgrounds, which carry inducible Xist on the *Castaneous* and *Domesticus* alleles respectively. ChIP-seq performed on YY1-FKBP12$^{F36V}$ lines using an antibody raised against the T7 epitope generated data with extremely low-background signal compared to anti-YY1 antibodies (Dataset EV2). This enabled quantitative allele-specific analysis of YY1 binding for any peaks which overlap strain-specific SNPs (Fig. 3B) over time courses of XCI. The binding of YY1 to target sites on Xi progressively decreases upon Xist induction with NPC differentiation (Figs. 3C and EV2B), but at a slower rate than both

**Figure 2.   Persistently accessible REs and slow-silencing genes are associated with binding of the transcription factor YY1.**

(A) Classification of equal-sizes groups of "depleted" and "persistent" ChrX REs. The group of "persistent" REs includes both REs with a halftime slower than the arbitrary threshold ($n = 5.4$ days) and peaks which remain biaccessible in iXist-ChrX$_{Dom}$ NPCs (allelic ratio > 0.2). (B) HOMER (http://homer.ucsd.edu/homer/) results of the top 2 TF motifs significantly enriched in "persistent" REs compared to "depleted" REs (for full results, see Dataset EV1). (C) Above: Consensus YY1 peaks from four replicates of ChIP-seq performed in iXist-ChrX mESCs with an anti-YY1 antibody ($n = 2$ iXist-ChrX$_{Cast}$, $n = 2$ iXist-ChrX$_{Dom}$). The numbers of YY1 peaks on ChrX reported here are prior to applying filters for allelic analysis (see "Methods"). Below: YY1 ChIP-seq genome track showing strong YY1 enrichment at the promoter of the slow-silencing gene *Hcfc1* and an upstream distal RE (track is an average of all 4 replicates). (D) Pie charts showing the proportions of depleted and persistent REs which overlap with a ChIP-seq peak of YY1 binding. (E) Boxplot demonstrating slower loss of accessibility for YY1-binding REs, quantified for all REs for which it is possible to calculate accessibility halftimes (657/790). Boxes span first to third quartiles with a central line indicating the median. Whiskers extend to 1.5 * the interquartile range and outliers outside this range are plotted as separate points. Significance (*P* value) calculated by Wilcoxon rank-sum test. Numbers of REs in each category are indicated below. (F) Above: Numbers of autosomal and X-linked genes classified as "direct YY1 targets" by a the presence of a YY1 ChIP-seq peak directly overlapping their TSS. Only genes which meet minimal thresholds for allelic expression (i.e., are sufficiently expressed in iXist-ChrX mESCs) were considered. Below: Pie charts showing the proportion of YY1 targets within groups of fast, intermediate and slow-silencing genes defined from ChrRNA-seq silencing analysis in iXist-ChrX$_{Cast}$ (Bowness et al, 2022). (G) Boxplots comparing silencing halftimes of YY1 targets versus non-target genes in iXist-ChrX$_{Cast}$, separated on the left by initial gene expression categories. Boxes span first to third quartiles with a central line indicating the median. Whiskers extend to 1.5 * the interquartile range and outliers outside this range are plotted as separate points. Significance (*P* values) calculated by Wilcoxon rank-sum tests. Numbers of genes compared are indicated below each box. (H) Boxplot showing greater residual Xi expression of YY1 target genes after 10 days of Xist induction in mESCs (iXist-ChrX$_{Cast}$ ChrRNA-seq data from GSE185852). Boxes span first to third quartiles with a central line indicating the median. Whiskers extend to 1.5 * the interquartile range and outliers outside this range are plotted as separate points. Significance (*P* value) calculated by Wilcoxon rank-sum test. The numbers of genes in each category are indicated below.

gene silencing (Fig. 3D) and chromatin accessibility (Fig. EV2C), and with notable exceptions such as the clusters of YY1-binding sites at *Firre* and *Dxz4* loci which become more enriched on Xi (Fig. EV2D). Cells induced with doxycycline but maintained in mESC culture conditions retain significantly more YY1 binding on Xi than cells differentiated towards NPCs (Figs. 3C and EV2B), implying that cellular differentiation is required for complete eviction of YY1, and is consistent with our previous observation that gene silencing does not progress beyond a certain point in mESCs (Bowness et al, 2022).

To further assess the degree to which YY1 binding persists on Xi we used allelic ChIP-seq to compare the retention of YY1 binding at sites on ChrX with that of a different arbitrary TF, OCT4. Following 6 days of Xist induction and chromosome silencing in mESCs, OCT4 binding is reduced at target sites on Xi (Fig. 3E) to a greater degree than YY1 binding at the same timepoint (Fig. 3F). This observation suggests that YY1 may have unique characteristics which enable binding to Xi chromatin until later events of Xist-mediated chromatin inactivation, although we did not extend the comparison by performing ChIP-seq for other transcription factors.

In a final set of experiments, we tested the hypothesis that the association between YY1 binding and slow-silencing genes may indicate a direct role for YY1 in impeding Xist-mediated silencing at its target genes. To investigate this, we made use of the FKBP12$^{F36V}$ degron tag in the engineered cell lines to acutely degrade YY1 and then measured the consequences on Xist-mediated silencing by ChrRNA-seq. In these lines, complete degradation of YY1-FKBP12$^{F36V}$ occurs upon treating cells with 100 nM dTAG-13 (Fig. 4A), and mESCs can be maintained in culture without YY1 for 3–5 days before reduced viability becomes evident. With this mind, we chose to apply dTAG treatment for 52 h and harvested samples for ChrRNA-seq at three timepoints of Xist induction in mESCs - 0 (uninduced), 2 and 6 days (Fig. 4B). We selected the day 2 timepoint, in which YY1 degradation precedes Xist induction, to investigate the role of YY1 in both initiation and establishment of Xist-mediated silencing. By contrast, in the day 6 timepoint YY1 degradation is triggered after almost 4 days of Xist induction in mESCs, by which time gene silencing has reached its maximum but slow-silencing genes are

residually expressed from Xi in the absence of cellular differentiation (Bowness et al, 2022). Thus, we chose this timepoint to assess if removal of YY1 increases the repression of genes at a later stage of the XCI process. Xist RNA-FISH analysis at both day 2 and day 6 timepoints indicated normal localisation over the X chromosome territory (Fig. EV3A), contrasting with studies in other cellular models suggesting roles for YY1 in Xist RNA spreading (Jeon and Lee, 2011) and/or localisation (Wang et al, 2016).

Upon YY1 degradation in uninduced cells (day 0), widespread gene expression changes occur in both directions, with direct targets more likely to decrease in expression after YY1 removal (Fig. EV3B). This broadly agrees with the etymology of YY1 (Yin Yang 1) as having either co-activating or co-repressive effects in a context-dependent manner (Shi et al, 1991; Park and Atchison, 1991) as well as literature reporting closer associations with coactivators such as P300, INOV080, BAF and Mediator complexes in mammalian cells (Yao et al, 2001; Cai et al, 2007; Wang et al, 2018b; Beagan et al, 2017). However, at the timepoint measured in this study (52 h) we cannot distinguish direct YY1 regulation from indirect effects of YY1 degradation. A previous study characterising YY1-FKBP mESCs reported similar gene expression changes in both directions upon 24 h of YY1 depletion via dTAG treatment (Weintraub et al, 2017).

In both 2-day and 6-day induced iXist-ChrX cells, dTAG treatment leads to a more skewed (i.e., lower) allelic ratio (Fig. 4C), demonstrating increased gene silencing upon degradation of YY1. Given the widespread gene expression effects upon loss of YY1, it was important to investigate if this effect on allelic ratio could be accounted for by gene expression changes occurring on Xa. Analysis of median log2-fold changes for each allele separately shows that the average gene expression fold change on Xa is close to 0 (Fig. 4D). By contrast, expression of genes on Xi skew negatively in dTAG-treated samples, signifying an increase in their degree of silencing. This trend is also clear upon inspection of X-linked genes individually (Figs. 4E and EV3C), as for example expression of *Hcfc1* (the X-linked gene with the strongest peak of YY1 binding at its promoter) increases from Xa upon YY1 degradation, but decreases from Xi. In general, larger increases in silencing occur at direct YY1 target genes (Fig. 4F), suggesting that

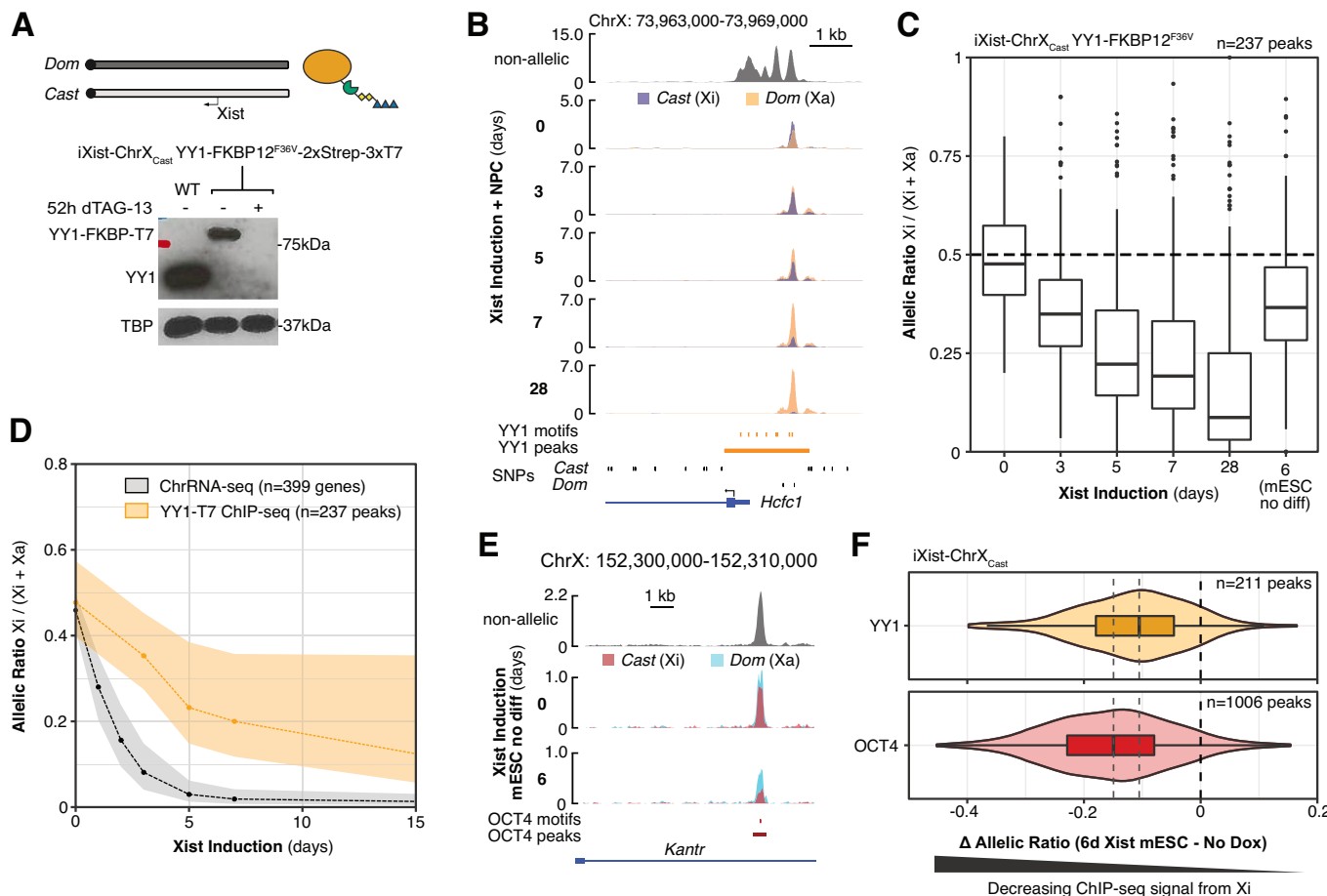

**Figure 3. YY1 binding is lost slowly from Xi during XCI.**

(A) Above: Schematic of the iXist-ChrX$_{Cast}$ YY1-FKBP12$^{F36V}$-2xStrep-3xT7 cell line, which allows for both allelic ChIP-seq and acute degradation of endogenous YY1 protein in the context of inducible XCI. An equivalent line engineered in the reciprocal allelic background (iXist-ChrX$_{Dom}$) is shown in Fig. EV2A. Below: Western blot of YY1-FKBP12$^{F36V}$-2xStrep-3xT7 fusion protein expression and degradation upon addition of 100 nM dTAG-13 (iXist-ChrX$_{Cast}$ background). TBP acts as a loading control. (B) Genome track of YY1-T7 ChIP-seq, showing the decrease in Xi-specific YY1 binding at the *Hcfc1* promoter over a time course of Xist induction with NPC differentiation. (C) Boxplot of decreasing YY1 binding on Xi, as measured by the allelic ratio of reads in each peak, over a time course of Xist induction with NPC differentiation. Boxes span first to third quartiles with a central line indicating the median. Whiskers extend to 1.5 * the interquartile range and outliers outside this range are plotted as separate points. (D) Ribbon plot comparing the dynamics of decreasing YY1 binding with gene silencing in the parental iXist-ChrX$_{Cast}$ line. Dashed lines connect median averages for each timepoint and shaded areas trace interquartile ranges. (E) Genome track of OCT4 ChIP-seq showing the decrease in Xi-specific OCT4 binding at an intronic peak in *Kantr* after 6 days of Xist induction in mESCs. (F) Violin plot comparing the decrease in Xi binding between YY1 and OCT4 after 6 days of Xist induction. For each peak of TF binding, loss of ChIP-seq signal from Xi is calculated as the change in allelic ratio from day 0 to day 6, which decreases by a median of 0.103 for YY1 and 0.149 for OCT4 (indicated by dashed lines). Boxes span first to third quartiles with whiskers extending to 1.5 * the interquartile range and a central median line. Violin width indicates a kernel density estimation of the data distribution.

YY1 degradation removes barriers to silencing pathways primarily at the level of individual loci. A previous study showed that YY1 binding to the major *Xist* enhancer in exon 1 promotes Xist RNA expression (Makhlouf et al, 2014). Our ChrRNA-seq analysis in 2-day and 6-day induced iXist-ChrX mESCs did not reveal a decrease in Xist RNA levels upon YY1 depletion, with levels if anything being slightly elevated (Figs. 4G and EV3D). We presume this difference is related to using the inducible TetOn promoter in place of the native *Xist* promoter. These elevated levels of Xist upon loss of YY1, attributable to either increased expression or greater RNA stability, may also contribute to the observed increase in X-linked gene silencing.

In summary, we set out to use ATAC-seq to measure loss of chromatin accessibility from REs – and as a proxy to monitor

progressive eviction of TFs – over a time course of several days following induction of Xist-mediated silencing in differentiating XX mESCs. It is well documented that genomic location relative to the *Xist* locus (Marks et al, 2015; Barros de Andrade E Sousa et al, 2019) or preferred Xist RNA association sites (Markaki et al, 2021), as well as initial levels of expression (Nesterova et al, 2019), play major roles in determining the silencing kinetics of different X-linked genes. Our analysis showed that the rate accessibility reduction from Xi REs also strongly correlates with the silencing kinetics of associated genes, and revealed enrichment of the TF YY1 at persistently accessible promoters and enhancers and at slow-silencing genes. In further experiments, we demonstrated that YY1 binding is retained on Xi chromatin to a greater extent than other TFs, including in a direct comparison, OCT4, and that YY1

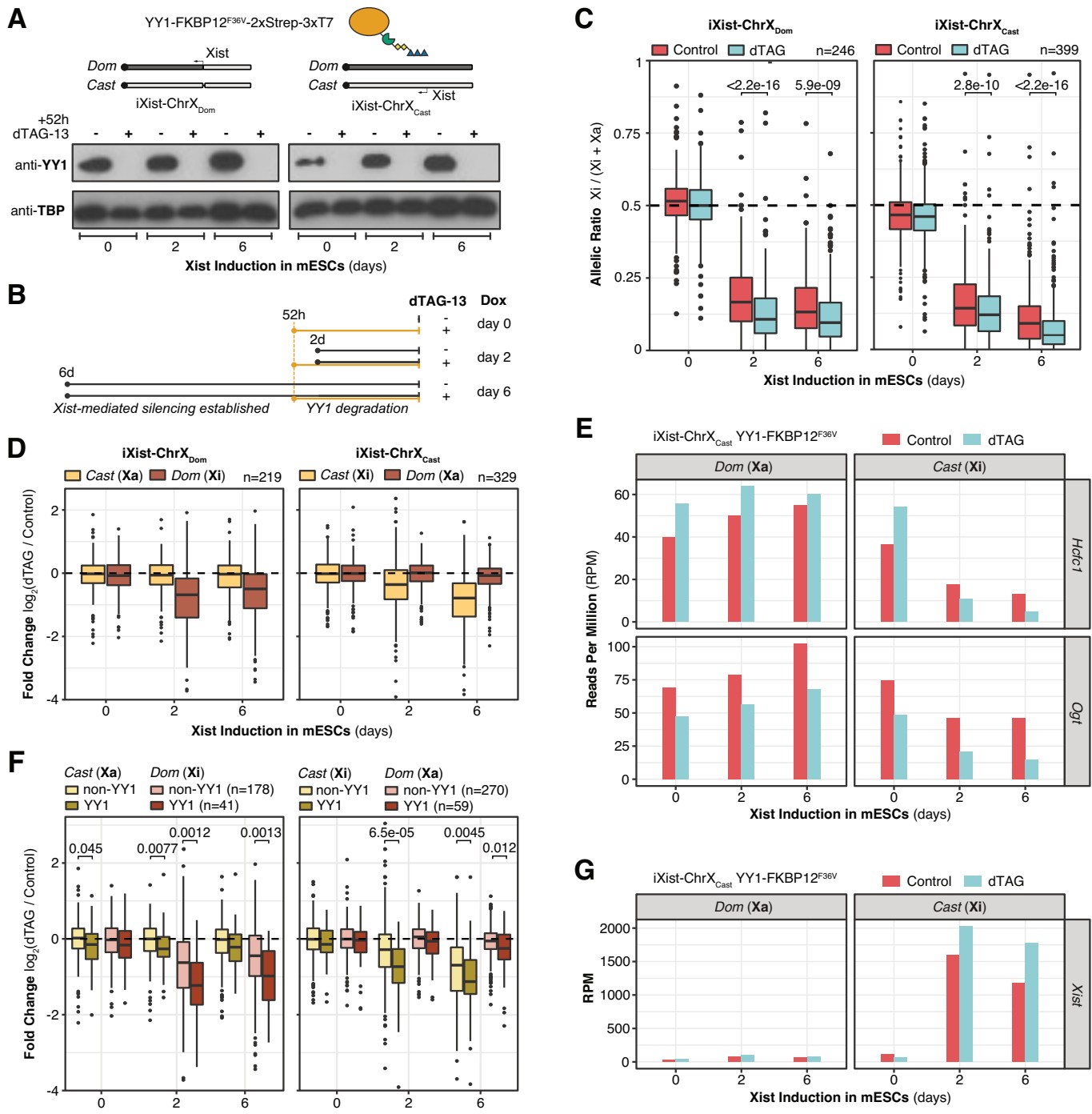

depletion leads to increased Xi silencing, particularly at YY1 target genes. Taken together, these results establish YY1 binding as an additional gene-intrinsic determinant of slow silencing, implying that YY1 acts as a barrier that must be removed for genes on Xi to fully inactivate.

Our observations also indicate that complete YY1 eviction requires late steps in the X inactivation process associated with cellular differentiation. A candidate for mediating eventual eviction is the chromosomal protein SmcHD1, which is enriched on Xi only

after several days of differentiation (Gendrel et al, 2012; Bowness et al, 2022). Consistent with this hypothesis, YY1-binding REs remain markedly more accessible on Xi in ATAC-seq data of SmcHD1 knock-out (KO) cell lines which fail to completely establish XCI when differentiated to NPCs (Fig. EV4) (for the gene silencing defect in SmcHD1 KO see Fig. 6 in Bowness et al, 2022). SmcHD1 is thought to function by increasing the compaction of Xi (Wang et al, 2018a; Jansz et al, 2018; Gdula et al, 2019), but also promotes the accumulation of H3K9me3 (Ichihara et al, 2022) and

**Figure 4.  Degradation of YY1 leads to increased silencing of Xi in mESCs without differentiation.**

(A) Above: Schematic of the YY1-FKBP12$^{F36V}$-2xStrep-3xT7 cell lines derived in both iXist-ChrX$_{Dom}$ and iXist-ChrX$_{Cast}$ backgrounds. Below: Western blots of YY1 fusion protein degradation upon addition of 100 nM dTAG-13. TBP acts as a loading control. (B) Schematic of the design of ChrRNA-seq experiments investigating the consequences of YY1 degradation at three timepoints of Xist-mediated silencing: day 0 (irrespective of XCI), day 2 (prior to Xist induction), and day 6 (after establishment of silencing). (C) Boxplots of allelic ChrRNA-seq analysis showing increased gene silencing upon YY1 degradation in iXist-ChrX$_{Dom}$ (left, $n = 246$ genes) and iXist-ChrX$_{Cast}$ (right, $n = 399$ genes) mESC lines. Boxes span first to third quartiles with a central line indicating the median. Whiskers extend to 1.5 * the interquartile range and outliers outside this range are plotted as separate points. Significance comparisons ($P$ values) calculated by paired $T$ tests. (D) Boxplots showing fold changes in allelic expression of X-linked genes between dTAG-treated and control samples. On average, expression from Xi decreases (indicating increased silencing) whereas expression from Xa and both alleles of uninduced mESCs remains unchanged. Boxes span first to third quartiles with a central line indicating the median. Whiskers extend to 1.5 * the interquartile range and outliers outside this range are plotted as separate points. The numbers of genes in each line that it is possible to calculate fold changes for are displayed above the plots. (E) Relative allele-separated expression of two example YY1 target genes, *Hcfc1* and *Ogt*, in ChrRNA-seq experiments performed in the iXist-ChrX$_{Cast}$ line. (F) As (D) but displaying YY1 target genes and non-YY1 genes separately, with the number of genes in each category given above. Significant differences ($P$ values) in Xi expression fold changes were calculated by Wilcoxon rank-sum tests. Other comparisons for Xa are non-significant ($P > 0.05$). (G) Levels of chromatin-associated Xist RNA in YY1-degron ChrRNA-seq experiments performed in the iXist-ChrX$_{Cast}$ line.

CpG island DNA methylation (Blewitt et al, 2008; Gendrel et al, 2012). The latter pathway may be of particular relevance as YY1 binding is blocked by CpG methylation (Kim et al, 2003; Makhlouf et al, 2014; Fang et al, 2019b).

Finally, although highly efficient silencing mediated by inducible Xist means that our iXist-ChrX cellular model is ill-suited to the study of escapees, YY1 binding has been reported to be associated with genes that escape XCI in humans (Chen et al, 2016). We note the overlap between slow-silencing genes in stem cell models and facultative escapees in other mouse and human tissues (Berletch et al, 2015; Tukiainen et al, 2017), and thus suggest YY1 as an interesting candidate for further study in contexts where genes escaping XCI have translational implications for sex-biased human diseases (Dunford et al, 2017; Souyris et al, 2018).

## Methods

### Mouse embryonic stem cell (mESC) culture

Female (XX) iXist-ChrX mESC lines containing doxycycline-inducible Xist (iXist-ChrX) have been described previously (Nesterova et al, 2019) and were originally derived from the parental F1 2-1 hybrid line (129/Sv-Cast/Ei, a gift from J Gribnau).

mESCs were maintained in Dulbecco's Modified Eagle Medium (DMEM; Life Technologies) with 10% foetal calf serum (FCS; ThermoFisher), 2 mM L-glutamine, 0.1 mM non-essential amino acids, 50 μM β-mercaptoethanol, 100 U/mL penicillin/100 μg/mL streptomycin (all from Life Technologies) and 1000 U/mL LIF (made in-house). Cells were grown on gelatin-coated plates under standard cell culture conditions (37 °C, 5% CO$_2$, humid) on a "feeder" layer of SNLP mouse fibroblasts mitotically inactivated by Mitomycin-C (Sigma Aldrich). mESCs were routinely passaged at ~80% confluency every 2–3 days using TrypLE Express (Thermo-Fisher) and tested for mycoplasma contamination. Before experiments, feeder cells were removed by pre-plating to gelatinised cell culture dishes for 30–40 min, with slower-attaching mESCs taken from suspension and plated to new gelatinised dishes.

Xist expression from the TetOn promoter was induced by the addition of 1 μg/ml doxycycline (Sigma, D9891). dTAG-13 (Tocris, #6605) was applied to FKBP12$^{F36V}$-fusion cell lines at 100 nM to induce protein degradation.

## mESC to neural progenitor cell (NPC) differentiation

ES to NPC differentiation was performed according to standard protocols with small modifications as previously described (Conti et al, 2005; Splinter et al, 2011; Bowness et al, 2022). First, mESCs were extensively isolated from feeder cells by four rounds of pre-plating, each for 35–40 min. $0.5 \times 10^6$ mESCs were plated to gelatin-coated T25 flasks in N2B27 media (50:50 DMEM/F-12:Neurobasal (Gibco) supplemented with 1× N2 and 1× B27 (ThermoFisher), 1 mM L-glutamine, 100 μM β-mercaptoethanol, 50 U/mL penicillin/50 μg/mL streptomycin (all from Life Technologies) with 1 μg/ml doxycycline for continuous Xist induction. Media changes were performed every day except day 1. After 7 days, cells were detached with Accutase (Merck Life Sciences) and $3 \times 10^6$ cells were plated to grow in suspension in 90 mm bacterial petri dishes containing N2B27 media supplemented with 1 μg/ml doxycycline and 10 ng/ml EGF and FGF (Peprotech). Embryoid-body-like aggregates were collected by mild centrifugation ($100 \times g$ for 2 min) on day 10 and re-plated to 90 mm gelatinised cell culture dishes in N2B27 + dox + FGF/EGF media, with media changes performed every other day. Upon reaching ~80% confluency, NPC outgrowths were passaged using Accutase at a 1:3–1:4 ratio to new cell culture dishes until the establishment of a homogenous cellular population of NPCs from approximately day 15–17.

## Derivation of YY1-FKBP12$^{F36V}$-3xT7 cell lines

YY1-FKBP12$^{F36V}$-3xT7 cell lines were engineered from iXist-ChrX cell lines by CRISPR/Cas9-mediated homologous recombination. Single guide RNAs for introducing double-strand breaks at the C-terminal region of *Yy1* were designed using the CRISPOR online tool (Concordet and Haeussler, 2018) and cloned into the pX459 plasmid background (Addgene #62988) using the Zhang lab digestion-ligation protocol (Broad Institute). The homology vector was cloned by Gibson assembly (Gibson Assembly Master Mix kit, NEB) using primers given in Dataset EV3 into a restriction-enzyme digested pCAG backbone plasmid. Homology sequences of 600–900 bp flanking the cut site of C-terminal *Yy1* were amplified by PCR from iXist-ChrX genomic DNA using FastStart High Fidelity enzyme (Sigma Aldrich). DNA sequence encoding the C-terminal 3xT7-2xStrepII-FKBP12$^{F36V}$ tag was amplified from a plasmid gifted by R. Klose lab (Dimitrova et al, 2022).

Twenty-four hours prior to transfection, $1–1.5 \times 10^6$ mESCs were plated to wells of a six-well plate, and Pen/Strep was removed from the growth media ~2 h prior to transfection. Cells were then co-transfected with the two plasmids at a molar ratio of 6:1 (2.5 µg of homology vector, ~0.7 µg of sgRNA vector) using Lipofectamine2000 (ThermoFisher). The next day, each well was split to 10 mm plates at low density and cells were treated with puromycin selection (3 µg/mL for iXist-ChrX$_{Dom}$, 5 µg/mL for iXist-ChrX$_{Cast}$) from 48 to 96 h post-transfection. Following wash out of puromycin, cells were grown under mESC culture conditions for 8–10 days until clonal colonies could be isolated to wells of 96-well plates and positive clones selected and expanded. Multiple clones for YY1-FKBP12$^{F36V}$-3xT7 cell lines in both parental iXist-ChrX backgrounds were screened and validated by PCR and western blot (see below). Clones aF1 (iXist-ChrX$_{Dom}$) and cC3 (iXist-ChrX$_{Cast}$) were chosen for further experiments. The YY1-FKBP12$^{F36V}$-3xT7 homology plasmid and mESC lines derived in this study are available upon request.

## Western blotting

To make total protein extract, a pellet of $\sim 1 \times 10^7$ cells was resuspended in 100 µl 1× RIPA buffer (Cell Signaling, #9806) with 1× PIC (cOmplete EDTA-free, Merck), 0.5 µL/ml Benzonase (Merck, E8263-5KU) and 5 mM PMSF and placed in a shaker for 30 min at room temperature. Extracts were then snap-frozen and stored at −80 °C. Prior to performing western blots, samples were quantified by Bradford's assay (Bio-Rad). For each sample, 20 µg protein resuspended in SMASH buffer (50 mM Tris-HCl pH 6.8, 10% glycerol, 2% SDS, 0.02% Bromophenol Blue, 1% β-mercaptoethanol) was loaded onto home-made polyacrylamide gels for electrophoresis (90 V 30 min, 180 V 60 min) and transferred to PVDF membranes using the Trans-blot Turbo (Bio/Rad) "Mixed Mw" setting. Membranes were blocked for 1 h at room temperature in blocking buffer of 5% milk powder (Marvel) dissolved in TBST (100 mM Tris pH 7.5, 0.9% NaCl, 0.1% Tween). Blots were incubated with primary anti-YY1 antibody (Santa Cruz, sc-7431) or loading control anti-TBP antibody (Abcam, #51841) overnight at 4 °C, washed four times with blocking buffer, then incubated on rollers at room temperature for 1 h with an anti-mouse secondary antibody conjugated to horseradish peroxidase (VWR, NXA931). After four further washes for 10 min each (2× blocking buffer, 1× TBST, 1× PBS), membranes were developed using Clarity Western ECL substrate (Bio-Rad).

## Xist RNA-FISH

iXist-ChrX YY1-FKBP12$^{F36V}$-3xT7 cells for each condition were grown on gelatin-coated 22-mm coverslips in wells of six-well plates and fixed at 60–70% confluency. Xist expression was induced either 2 or 6 days prior to fixation, with dTAG-13 added to degrade YY1 52 h prior to fixation. At collection, cells on coverslips were washed once with PBS, fixed in the six-well plate with 3% formaldehyde (pH 7) for 10 min, then washed once with PBS, twice with PBST.5 (0.05% Tween-20 in PBS), and transferred into a new six-well dish for permeabilization in 0.2% Triton X-100 in PBS for 10 min at RT. After three further PBST.5 washes, cells on coverslips were subjected to ethanol dehydration by an initial incubation with 70% EtOH (for 30 min at RT), then progressive

exchanges to 80%, 90% and finally 100% EtOH. Xist FISH probe was prepared from an 18 kb cloned cDNA (pBS_Xist) spanning the whole Xist transcript using a Nick translation kit (Enzo Biosciences). The FISH hybridisation mix consisted of: 3 µL Texas Red-labelled Xist probe (~50 ng DNA), 1 µL 10 mg/mL Salmon Sperm DNA, 0.4 µL 3 M NaOAc and 3 volumes of 100% EtOH per sample. This was precipitated by centrifugation ($20,000 \times g$ for 20 min at 4 °C), washed with 70% EtOH, air-dried, then resuspended in 6 µL deionised formamide (Merck Life Science) per hybridisation and incubated in a shaker (1400 rpm) at 42 °C for at least 30 min. 2× hybridisation buffer (4× SSC, 20% dextran sulphate, 2 mg/mL BSA (NEB), 1/10 volume nuclease-free water and 1/10 volume vanadyl-ribonucleoside complex (VRC; prewarmed at 65 °C for 5 min before use)) was denatured at 75 °C for 5 min, placed back on ice to cool, then mixed with hybridisation mix. Each coverslip was hybridised with 30 µL probe/hybridisation mix in a humid box at 37 °C overnight. The next day, coverslips were washed three times for 5 min at 42 °C with pre-warmed 50% formamide/2× saline-sodium citrate buffer (1/10 20× SSC in PBST.5), then subjected to further washes (3× 2× SSC, 1× PBST.5, 1× PBS, each for 5 min using a 42 °C hot plate) before being mounted with VECTASHIELD plus DAPI (Vector Labs) onto Superfrost Plus microscopy slides (VWR). Slides were dried and sealed using clear nail polish for imaging.

Blinded RNA-FISH imaging was not performed, but multiple individuals inspected and collected images of the experimental slides. 5–10 images per sample were acquired with AxioVision software on an inverted fluorescence Axio Observer Z.1 microscope (Zeiss) using a PlanApo ×63/1.4 NA oil-immersion objective. For each field of view, a variable number of Z-stacks were selected for imaging (10–25), intended to ensure that the Xist RNA territories of every cell were in focus in at least one Z-plane. Maximum-Intensity Z-projections were generated for all images using a custom bulk processing Fiji macro "Zproject.ijm". Scale bars were also calculated in Fiji (Schindelin et al, 2012). This macro is available with all raw images at BioImage Archive S-BIAD1091.

## Assay for transpose-accessible chromatin with sequencing (ATAC-seq)

Chromatin accessibility was assayed by ATAC-seq immediately upon collection using a protocol adapted from "Omni-ATAC" (Corces et al, 2017). Briefly, $1–5 \times 10^6$ cells were harvested as pellets, washed with PBS, and nuclei were isolated by incubation for 1 min at room temperature in 600 µl HS Lysis buffer (50 mM KCl, 10 mM MgSO$_4$.7H$_2$O, 5 mM HEPES, 0.05% NP40 (IGEPAL CA630), 1 mM PMSF, 3 mM DTT). Nuclei were then centrifuged at $1200 \times g$ for 5 min at 4 °C, followed by three washes with cold RSB buffer (10 mM NaCl, 10 mM Tris pH 7.4, 3 mM MgCl$_2$). After counting, $5 \times 10^5$ nuclei were centrifuged ($1500 \times g$ for 5 min at 4 °C) and resuspended in 50 µl H$_2$O. In total, $5 \times 10^4$ nuclei (5 µl) were taken for each transposition assay, performed in technical duplicate for each sample in a 50 µl transposition mix of: 1× Tn5 reaction buffer (10 mM TAPS, 5 mM MgCl$_2$, 10% dimethylformamide), 0.1% Tween-20 (Sigma), 0.01% Digitonin (Promega), 2.5 µl Tagment DNA TDE1 enzyme (Illumina), 16.5 µl PBS and 5 µl H$_2$O. As controls for transposition and mapping bias, tn5-digested "input" controls were made by performing tagmentation for 50 ng iXist-ChrX genomic DNA by a basic 50 µl transposition mix of 1X

TDE buffer and 2.5 µl TDE1 Enzyme (Illumina) in $H_2O$. Both sample and input mixes were incubated at 37 °C for 35 min in a thermomixer at 1000 rpm, then cleaned-up with ChIP DNA Clean and Concentrator kit (Zymo) and eluted in 14 µl elution buffer for storage at −20 °C. ATAC-seq libraries were prepared by ~8 cycles of PCR using NEBNext High Fidelity 2× PCR Master Mix (NEB) and primers containing Illumina barcodes (Buenrostro et al, 2013). Libraries were purified and size-selected using Agencourt AMPure XP bead clean-up (Beckman Coulter) to ensure a size distribution containing mono-, di- and tri-nucleosomal fragments between 150 and 800 bp.

## Chromatin immunoprecipitation of YY1 with sequencing (YY1 ChIP-seq)

For ChIP of YY1, $2–3 \times 10^7$ cells were collected and immediately processed for fixation using reagents provided in the truChIP Chromatin Shearing Kit with Formaldehyde (Covaris). After one wash with PBS, cells were resuspended with Fixing Buffer A and fixed with fresh 1% formaldehyde for exactly 10 min on a rotator at room temperature. After quenching and two washes with cold PBS, pellets of $1 \times 10^7$ were processed immediately or snap-frozen and stored at −80 °C. Nuclear and Chromatin extraction for sonication was performed according to the recommended protocol for "High Cell" numbers ($1 \times 10^7$ cells) in the Covaris truChIP Chromatin Shearing Kit manual (PN 010179). Chromatin was then sheared using a Covaris S220 Focused-ultrasonicator in AFA milliTUBEs with 1 ml of cells in Shearing Buffer D3. Typically, 20 min of sonication was required to achieve chromatin fragments of the desired target size (200–700 bp), and it was necessary to check chromatin shearing efficiency for each sample individually by agarose gel electrophoresis before proceeding with immunoprecipitation. After achieving satisfactory shearing, sonicated chromatin was adjusted to a final concentration of 150 mM NaCl and 1% Triton and pre-cleared by centrifugation at $10,000 \times g$ for 5 min at 4 °C.

Protein-A magnetic Dynabeads (40 µl per sample) were blocked in Chromatin IP buffer (Shearing Buffer D3 diluted with 2× Covaris truChIP IP Buffer and supplemented with 1× Protease Inhibitor Cocktail) with 1 mg/ml BSA and 1 mg/ml yeast tRNA for 1 h rotating at 4 °C. After two washes with IP buffer, blocked beads were added to chromatin extract for rotation for 1 h at 4 °C and then placed on a magnet. Supernatant chromatin was taken to a new Protein LoBind Eppendorf tube (total volume 1100 µl) and beads were discarded. At this step, 100 µl was taken to a new tube as a 10% input sample. Antibody was added to the remaining 1 ml extract (5 µl anti-T7 (Cell Signaling D9E1X) or 5 µl anti-YY1 (Cell Signaling D5D9Z)) and incubated on a rotator at 4 °C overnight. The following day, blocked Protein-A magnetic Dynabeads were added to each tube of chromatin and rotated for 1 h at 4 °C to allow for binding of Ab-conjugated chromatin fragments to beads. Beads were washed in buffers of increasing salt as follows: 2× Low Salt (0.1% SDS, 1% Triton X-100, 2 mM EDTA, 20 mM Tris-HCl pH 8.1, 150 mM NaCl), 2× High Salt (0.1% SDS, 1% Triton X-100, 2 mM EDTA, 20 mM Tris-HCl pH 8.1, 500 mM NaCl), 2× LiCl (0.25 M LiCl, 1% NP40, 1% Deoxycholate, 1 mM EDTA, 10 mM Tris-HCl pH 8.1), 2× TE buffer. Each wash was performed by rotating the beads for 3 min at 4 °C before placing samples back on a magnetic rack and discarding the supernatant. After washes,

chromatin fragments were extracted by two rounds of elution from beads (each of 15 min at room temperature) in a total of 120 µl fresh elution buffer (1% SDS, 0.1 M $NaHCO_3$). Reverse crosslinking of both eluted ChIP samples and inputs was performed by the addition of 1.2 µl 5 M NaCl and shaking incubation (1100 rpm) at 65 °C for 4 h, followed by treatment with RNaseA (1.2 µl of 1 mg/ml then shaking at 42 °C for 1 h) and ProteinaseK (1.2 µl of 20 mg/ml then shaking at 45 °C for 1 h). ChIP DNA was purified with the Zymo ChIP DNA Clean & Concentrator kit (Zymo Research). ChIP enrichment was confirmed by qPCR using primers described in Dataset EV3. Roughly 1 ng of ChIP DNA was taken for Illumina sequencing library preparation using the NEBNext Ultra II DNA Library Prep Kit with NEBNext Single indices (E7645) with 12 cycles of PCR amplification.

## ChIP-seq comparison of OCT4 and YY1

For the comparison between Xi binding of YY1 with OCT4, ChIP-seq was performed for both proteins from the same sample chromatin extracts, processed as follows:

In total, $5 \times 10^7$ mESCs were collected from confluent 150-mm dishes, washed once with PBS and pelleted by centrifugation at $400 \times g$ for 4 min. Cell pellets were resuspended for double-crosslinking in 10 ml 2 mM disuccinimidyl glutarate (DSG) followed by the addition of 1% formaldehyde and a further 12 min of incubation with mild rotation (at room temperature). Crosslinking was quenched by the addition of glycine to a final concentration of 135 µM and pellets were centrifuged ($400 \times g$ for 4 min at 4 °C) and washed once with cold PBS. Nuclei and chromatin extraction was then performed by sequential rounds of pellet resuspension, rotation at 4 °C for 10 min, and centrifugation ($400 \times g$ for 4 min at 4 °C) with LB1, LB2 and LB3 buffers: LB1 - 50 mM HEPES pH 7.9. 140 mM NaCl, 1 mM EDTA, 10% glycerol, 0.05% NP40 (IGEPAL CA630), 0.25% Triton X-100; LB2 - 10 mM Tris-HCl pH 8, 200 mM NaCl, 1 mM EDTA, 0.5 mM EGTA; LB3 - 10 mM Tris-HCl (pH 8.0), 200 mM NaCl, 1 mM EDTA, 0.5 mM EGTA, with freshly-added 0.1% sodium deoxycholate (Sigma) and 0.5% N-lauroylsarcosine (Sigma) (all buffers with freshly-added PIC). Crosslinked chromatin resuspended in 1 ml LB3 was then sonicated using a BioRuptor Pico (Diagenode) for 30 cycles of 30 s on/off. In total, 110 µl LB3 + 10% Triton X-100 (pre-warmed to 50 °C to help Triton dissolve) was added to 1 ml sonicated chromatin and mixed well by inversion. Max centrifugation ($16,000 \times g$ for 10 min at 4 °C) cleared debris to leave a chromatin supernatant. To verify successful sonication, an extract of chromatin (~50 µl) was taken for reverse cross-linking (200 mM NaCl solution in 65 °C shaker at 1000 rpm overnight) followed by RNase treatment (1 h 37 °C), ProteinaseK treatment (1 h 43 °C), and DNA extraction by DNA Clean and Concentrator kit5 (Zymo). DNA fragments were quantified by NanoDrop and verified to be of appropriate length (200–800 bp) by agarose gel electrophoresis.

For immunoprecipitation, ~150 µg of chromatin per IP (~100 µl of 1 ml) was diluted to 1.1 ml in ChIP-dilution buffer (1% Triton X-100, 1 mM EDTA, 20 mM Tris-HCl pH 8, 150 mM NaCl). Protein-A agarose beads (ThermoFisher) pre-blocked for 1 h with 0.2 mg/ml BSA and 50 µg/ml yeast tRNA were added to diluted chromatin (40 µl per 1 ml) and incubated for 30–60 min with rotation at 4 °C. Beads were pelleted by centrifugation ($1000 \times g$ for 4 min at 4 °C) and pre-cleared supernatant chromatin was taken to

new tubes for immunoprecipitation. Chromatin samples were then incubated overnight with 2.5 μg anti-OCT4A antibody (C3OA3C1 rabbit mAb, Cell Signaling) or anti-YY1 antibody (Cell Signaling D5D9Z) rotating at 4 °C, before blocked Protein-A agarose beads were again added and samples were places on a rotator for 1 h at 4 °C to bind antibody-bound chromatin fragments to beads. Agarose beads were then washed with low salt buffer (0.1% SDS, 1% Triton X-100, 2 mM EDTA, 20 mM Tris-HCl pH 8, 150 mM NaCl), high salt buffer (0.1% SDS, 1% Triton X-100, 2 mM EDTA, 20 mM Tris-HCl pH 8, 500 mM NaCl), LiCl buffer (0.25 M LiCl, 1% NP40, 1% sodium deoxycholate, 1 mM EDTA, 10 mM Tris-HCl pH 8) and two washes with TE buffer (10 mM Tris-HCl pH 8, 1 mM EDTA), with each ChIP wash consisting of rotation of beads for 3 min at 4 °C followed by centrifugation of beads at 1000× g for 2 min at 4 °C. After the washes, chromatin was eluted from beads for 30 min by rotation at room temperature in 150 μl fresh elution buffer (1% SDS, 0.1 M NaHCO3), followed by reverse cross-linking (as above) and DNA purification with ChIP DNA Clean and Concentrator kit (Zymo). Sequencing libraries were prepared from 1 to 5 ng ChIP DNA using the NEBNext Ultra II DNA Library Prep kit with NEBNext Single indices (E7645) and 14 final cycles of PCR amplification with the 65 °C elongation step reduced to 30 s. Final libraries were size-selected with AMPure XP beads to ensure a more optimal fragment size distribution for Illumina next-generation sequencing.

## Chromatin RNA-seq

In total, $2 \times 10^7$ mESCs were collected from confluent 90-mm dishes, washed once with PBS, then snap-frozen and stored at −80 °C. Chromatin extraction was performed as follows: Cell pellets were lysed on ice for 5 min in RLB (10 mM Tris pH 7.5, 10 mM KCl, 1.5 mM MgCl$_2$, and 0.1% NP40). Nuclei were then purified by centrifugation through 24% sucrose/RLB (2800 g for 10 min at 4 °C), resuspended in NUN1 (20 mM Tris pH 7.5, 75 mM NaCl, 0.5 mM EDTA, 50% glycerol, 0.1 mM DTT), and then lysed by gradual addition of an equal volume NUN2 (20 mM HEPES pH 7.9, 300 mM NaCl, 7.5 mM MgCl$_2$, 0.2 mM EDTA, 1 M Urea, 0.1 mM DTT). After 15 min incubation on ice with occasional vortexing, the chromatin fraction was isolated as the insoluble pellet after centrifugation (2800× g for 10 min at 4 °C). Chromatin pellets were resuspended in 1 mL TRIzol (Invitrogen) and fully homogenised and solubilised by eventually being passed through a 23-gauge needle ten times. This was followed by isolation of chromatin-associated RNA by TRIzol/chloroform extraction with isopropanol precipitation. Precipitated RNA pellets were washed twice with 70% ethanol and air-dried. Final ChRNA samples were then resuspended in H$_2$O, treated with TurboDNAse and measured by Nanodrop (both ThermoFisher). In all, 500 ng–1 μg of RNA was used for library preparation using the Illumina TruSeq stranded total RNA kit (RS-122-2301).

## Next-generation sequencing (NGS)

NGS libraries of ChRNA-seq, ATAC-seq, and ChIP-seq samples were loaded on a Bioanalyzer 2100 (Agilent) with High Sensitivity DNA chips to verify fragment size distribution between ~200 and 800 bp. If necessary, additional rounds of clean-up and/or size selection were performed using Agencourt AMPure XP beads

(Beckman Coulter) to remove residual adaptors or large (>1000 bp) fragments. Sample libraries were quantified using a Qubit fluorometer (Invitrogen) and pooled together. After pooling, final libraries were quantified by qPCR comparison to previously sequenced NGS libraries using Illumina P5/P7 qPCR primers and SensiMix SYBR (Bioline).

Paired-end sequencing was performed using an Illumina NextSeq500 (ChIP-seq - 2x81bp, 6 bp i7; ATAC-seq - 2x79bp, 8 bp i7).

## Allelic ATAC-seq analysis

Paired-end fastq files from ATAC-seq in iXist-ChrX$_{Dom}$ (Fig. 1) were mapped using bowtie2 (v2.3.5.1; (Langmead and Salzberg, 2012)) to a mm10 genome N-masked at positions of SNPs between *M. musculus domesticus* (129/Sv) or *M. musculus castaneous* (CAST) strains. The parameters "--very sensitive --no-discordant --no-mixed -X 2000" were used for mapping and any unmapped read pairs were removed. Aligned bam files were then sorted by samtools and PCR duplicates were marked and removed using the "MarkDuplicates" programme in the Picard toolkit (Broad Institute). For downstream allelic analysis, aligned reads were assigned to separate files corresponding to either the CAST ("genome1") or 129/Sv ("genome2") genomes by SNPsplit (v0.2.0; (Krueger and Andrews, 2016)) using the "--paired" parameter and a strain-specific SNP file compiled from UCSC annotations. Bigwig files of pileup tracks were generated by bamCoverage (deeptools v3.5.0; (Ramírez et al, 2016)) using the total library size as a normalisation scale factor. Tracks were loaded as Bigwig files to IGV (Robinson et al, 2011) or converted to bedGraphs for visualisation with SparK software (Kurtenbach and Harbour, 2019).

Quality of ATAC-seq libraries was assessed by the Transcription Start Site Enrichment (TSSE) score (https://www.encodeproject.org/data-standards/terms/). TSSE scores for each sample were calculated using the Bioconductor "ATAC-seqQC" package (Ou et al, 2018) and are given in Dataset EV4.

Peakcalling was performed on each replicate of ATAC-seq individually by MACS2 (v2.2.7.1; (Feng et al, 2012)) using parameters of "-f BAMPE -g mm -q 0.01". A custom R script using the "GenomicRanges" R package (Lawrence et al, 2013) was used to generate a consensus set of iXist-ChrX$_{Dom}$ mESC regulatory elements as regions covered by peaks in at least two sample replicates. This script also filters consensus peaks by lower and upper thresholds of 50 bp and 10 kb, respectively. Peaks were also called on tn5-digested gDNA samples by "-f BAMPE -g mm --broad --broad-cutoff 0.01" and these were subtracted from the consensus peak set as likely mapping artifacts.

Regulatory elements were assigned as "promoters" if they overlap within 500 bp of an NCBI RefSeq gene TSS, or otherwise as "distal". REs were classified as "CTCF-binding" if they overlap with a CTCF ChIP-seq peak — called from public data (GSM4776653; Hua et al, 2021) with parameters "-f BAMPE -g mm -q 0.01" — or "YY1-binding" if they overlap with a YY1 ChIP-seq peak (see below). In addition, REs were assigned to their nearest gene (TSS) by linear genomic proximity using bedtools closest.

For quantitative allelic analysis of RE accessibility over the time course of XCI, consensus ChrX REs were labelled and parsed into gtf file format by awk commands. These were used as input for featureCounts (Liao et al, 2014) to count sequencing fragments

overlapping REs in both the total and allele-specific alignment files for each timepoint. Only peaks containing at least 10 allelically assigned fragments in >80% of samples and showing biallelic signal in uninduced mESCs (0.15 < allelic ratio < 0.85) were retained ($n = 821$). Allelic ratios (Xi/(Xi + Xa)) were then calculated for all sample timepoints and used for further analysis.

For analysis of ATAC-seq time courses of iXist-ChrX WT versus SmcHD1 KO cell lines in Fig. EV4, consensus peak files were created for each set of ten samples for comparison (e.g., 5× iXist-ChrX$_{Dom}$ + 5 Xist-ChrX$_{Dom}$ SmcHD1 KO-A3) and annotated by overlap with YY1 ChIP-seq peaks. Count matrices were then generated by featureCounts and used for allelic ratio analysis. Filtering by minimum allelic fragments and initial allelic ratio was performed similarly to the original iXist-ChrX$_{Dom}$ time-course data, with fewer peaks passing allelic filters compared to Fig. 1 because of lower sequencing depth.

## Kinetic modelling of dynamic RE accessibility loss

Trajectories of decreasing allelic RE accessibility in the iXist-ChrX$_{Dom}$ ATAC-seq time course were fitted to curves of an exponential model using largely the same methodology as previously described for ChrRNA-seq data (Bowness et al, 2022). Specifically, each RE was fit using the "nlsLM" function of the "minpack.lm" R package (Elzhov et al, 2016) to a model of the form:

$$y = y_f + y_0 e^{-tk}$$

($y$ = allelic ratio; $t$ = time; $y_f$ = final allelic ratio; $y_0 + y_f$ = initial allelic ratio)

Unlike in our previous ChrRNA-seq silencing analysis, $y_f$ was not fixed at 0 for any peaks, as complete loss of Xi ATAC-seq signal from REs is rare. Halftimes of RE accessibility loss were calculated by the formula:

$$t_{1/2} = -\frac{1}{k}\ln\left(\frac{F(y_0 - y_f) - y_f}{y_0}\right)$$

Where $y_0$ and $y_f$ are parameters of the exponential model fit and $F = 0.5$ (to calculate half of the initial value). Overall, it was possible to calculate halftimes for 657 of the 821 chrX REs which passed initial filters for allelic analysis. Some REs demonstrate behaviours other than a progressive decrease in allelic ratio upon Xist induction (e.g., the CTCF sites at the *Firre* locus which increase in allelic ratio over XCI) and thus cannot be fitted with an exponential curve and a halftime value. 133/164 of these REs remain biallelically accessible in NPCs (Allelic Ratio > 0.20) and were retained for further analysis. These were classified together with REs with slow accessibility halftimes ($t_{1/2} > 5.4$ days) as "persistent REs" ($n = 395$). An equal-sized group of REs for which Xi accessibility rapidly decreases ($t_{1/2} < 5.4$ days) were classified as "depleted REs" ($n = 395$). These categories of "persistent" and "depleted" REs have similar properties in terms of sequence length (median bp: persistent = 991, depleted = 887) and initial

accessibility in mESCs (median day 0 ATAC enrichment score: persistent = 1133, depleted = 1176).

A table collating the halftimes and various classifications for all REs amenable to allelic analysis is provided as Dataset EV5.

## Motif enrichment and annotation

Motif analysis was performed by HOMER (Heinz et al, 2010) using the command "findMotifsGenome.pl" to search for motifs enriched in "persistent REs", using "depleted REs" as the background feature set ("-bg"). The enrichment results table (Dataset EV1) and figure (Fig. 2B) show the output of the "known" motif enrichment mode. YY1 motifs within RE sequences were annotated by FIMO from the MEME suite tools (Bailey et al, 2015), searching for the consensus mouse YY1 motif from the JASPAR database (YY1_MA0095.2; Castro-Mondragon et al, 2022).

## YY1 ChIP-seq analysis

Mapping and processing of YY1 ChIP-seq data was performed using a similar pipeline to ATAC-seq data (see above). A consensus set of YY1 ChIP-seq peaks in iXist-ChrX was generated using anti-YY1 ChIP-seq data generated in iXist-ChrX mESCs. Using the same methodology as ATAC-seq, peaks were called from the four samples (2× iXist-ChrX$_{Dom}$, 2× iXist-ChrX$_{Cast}$) individually with MACS2 "-f BAMPE -g mm -q 0.01", and regions where two or more peaks overlap were consolidated as the consensus peak set. A consensus input blacklist from peaks called in the four input samples (by MACS2 "-f BAMPE -g mm --broad --broad-cutoff 0.01") was also generated and subtracted from the consensus YY1 peaks. Peaks were also annotated with a YY1 "enrichment score" in mESCs using the "cov" command from the BAMscale tool (Pongor et al, 2020).

YY1 peaks were defined as "promoters" if they overlap within 500 bp of an NCBI RefSeq gene TSS, or otherwise as "distal". YY1 peaks were also assigned to their nearest gene (TSS) by bedtools closest, and in cases of direct overlap between a YY1 peak and TSS, the associated gene was defined as a "direct YY1 target" ($n = 2168$ genome-wide).

YY1 ChIP-seq performed in iXist-ChrX YY1-FKBP12$^{F36V}$-3xT7 lines using the anti-T7 antibody generated data with improved signal:noise compared to ChIP-seq using the endogenous anti-YY1 antibody. This is evident from superior FRiP (Fraction Reads in Peaks) scores (see Dataset EV2) in anti-T7 libraries using the consensus YY1 peak set (derived from anti-YY1 data)*. Accordingly, anti-T7 YY1 ChIP-seq was used for the quantitative allelic analysis of YY1 binding on Xi over time courses of XCI. Consensus YY1 peaks from YY1-T7 ChIP data were labelled, parsed into gtf file format, and used as input for featureCounts (Liao et al, 2014) to count sequencing fragments overlapping YY1 peaks in both the total and allele-specific alignment files for each timepoint. Counts matrices for YY1 time courses in both aF1 and cC3 lines were loaded to RStudio, whereupon allelic filters of >10 allelically assigned fragments in >80% of samples and biallelic signal in uninduced mESCs (0.2 < allelic ratio < 0.8) were applied to cC3 and aF1 datasets to result in $n = 237$ and $n = 195$ peaks respectively. Allelic ratios (Xi/(Xi + Xa)) were then calculated for all sample timepoints and used for further analysis.

*It is possible to call many more peaks from anti-T7 YY1 ChIP-seq data, however, these peaks are of smaller magnitude and rarely contain YY1 motifs, so we reasoned that these peaks likely reflect secondary enrichment of YY1 cofactors rather than direct sites of YY1 binding to DNA. As such, for the purposes of this analysis we opted to use peaks called from the anti-YY1 Ab data, which represents a smaller but higher confidence set of locations where YY1 binds in the genome.

## OCT4 ChIP-seq analysis

Mapping, processing and peakcalling of OCT4 ChIP-seq data was performed exactly as anti-YY1 ChIP-seq. Consensus OCT4 peaks were labelled, parsed into gtf file format, and used as input for featureCounts (Liao et al, 2014) to count sequencing fragments overlapping OCT4 peaks in both the total and allele-specific alignment files for d0 and d6 mESC timepoints. The same filtering criteria and calculations were performed in R for OCT4 peaks and YY1 peaks to maximise comparability of allelic ratio measurements between the two transcription factors, although the number of peaks on ChrX passing filtering criteria was greater for OCT4 ($n = 1005$) than YY1 ($n = 237$).

## ChrRNA-seq analysis

Further analysis of wild-type iXist-ChrX ChrRNA-seq time courses was not performed for this paper. Gene silencing halftimes and categories of genes (i.e., fast/intermediate/slow silencing or low/medium/high initial expression) were taken from a table in our previous publication, which summarises this information for all X-linked genes amenable to allelic ChrRNA-seq analysis in iXist-ChrX cell lines (Bowness et al, 2022). A version of this table updated to include classifications of genes as YY1 binding is provided as Dataset EV6. ChrRNA-seq data from iXist-ChrX cells induced for 10 days in mESC conditions without differentiation is available at GSE185852.

Mapping and processing of ChrRNA-seq data from iXist-ChrX YY1-FKBP12$^{F36V}$-3xT7 mESCs were performed by a similar pipeline to our previous studies (Nesterova et al, 2019; Bowness et al, 2022). Raw fastq files were first mapped to rRNA by bowtie2 (v2.3.5.1; Langmead and Salzberg, 2012), and rRNA-mapping reads were discarded. The remaining reads were aligned to the N-masked genome using STAR (v2.5.2b; Dobin et al, 2013) using parameters "-outFilterMultimapNmax 1 -outFilterMismatchNmax 4 -alignEndsType EndToEnd", and assigned to *Castaneous*/CAST or *Domesticus*/129 Sv genome files by SNPsplit (v0.2.0; Krueger and Andrews, 2016). For both "unsplit" and "allelic" files of each sample, read fragments overlapping genes were counted by featureCounts (Liao et al, 2014) "-t transcript -g gene id -s 2" using a non-redundant annotation file of all transcripts and lncRNAs from NCBI RefSeq.

Allelic analysis of gene silencing in aF1 and cC3 iXist-ChrX YY1-FKBP12$^{F36V}$-2xStrep-3xT7 mESCs was performed using R and RStudio on allelic count matrices produced by featureCounts. Genes which passed allelic filters in our previous analyses of wild-type gene silencing (i.e., included in Dataset EV6) were retained for Allelic Ratio analysis. Slightly fewer genes were amenable to Allelic Fold Change analysis (aF1 - 219/246; cC3 - 329/399) because fold changes cannot be calculated in situations where there are 0 counts for one allele of a gene (e.g., where the gene is completely silent on day 6).

Differential gene expression analysis between untreated and dTAG-13-treated samples was performed on "unsplit" counts matrices using DEseq2 (Love et al, 2014) using the "lfcShrink" transformation. aF1 and cC3 lines were evaluated separately, but all samples for each line (days 0, 2 and 6 of Xist induction) were grouped to test for genome-wide differential expression testing upon 52 h YY1 degradation. Volcano plots were generated using the *EnhancedVolcano* R package (Blighe et al, 2018).

## Quantification and statistical analysis

Statistical significance was calculated using base or "ggpubr" packages and each statistical test performed is stated in the relevant figure legends. All hypothesis tests were conducted in a two-sided manner. The Wilcoxon rank-sum test was favoured for comparisons between independent, unpaired samples (e.g., between two different categories of genes or REs). This test does not assume normal distributions and is a non-parametric alternative to the unpaired $T$ test. For comparing the same set of genes under different conditions, we elected to use the paired $T$ test. This parametric statistical test is relatively robust to deviations from normality in the distribution of differences between paired observations at the sample sizes compared in this study. All sample sizes ($n$) are provided and explained in figures and legends.

## Data availability

NGS datasets produced in this study are available in the GEO SuperSeries GSE240684. Raw RNA-FISH images are available in BioImage Archive S-BIAD1091. Records of all analysis performed in R for this study are hosted on GitHub at the following page: https://joebowness.github.io/YY1-XCI-analysis/.

## Peer review information

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

## Acknowledgements

The authors are grateful to members of the Brockdorff lab for critical discussion and suggestions. We would also like to thank Amanda Williams for loading and maintenance of the Illumina NextSeq500 machine and Oxford Biochemistry IT support for computer server maintenance. This work was funded by Wellcome grants 215513 (NB) and 203817 (JSB).

## Author contributions

**Joseph S Bowness**: Conceptualisation; Data curation; Formal analysis; Investigation; Methodology; Writing—original draft; Writing—review and editing. **Mafalda Almeida**: Investigation. **Tatyana B Nesterova**: Investigation. **Neil Brockdorff**: Conceptualisation; Supervision; Funding acquisition; Writing—original draft; Project administration; Writing—review and editing.

## Disclosure and competing interests statement

The authors declare no competing interests.

# Expanded View Figures

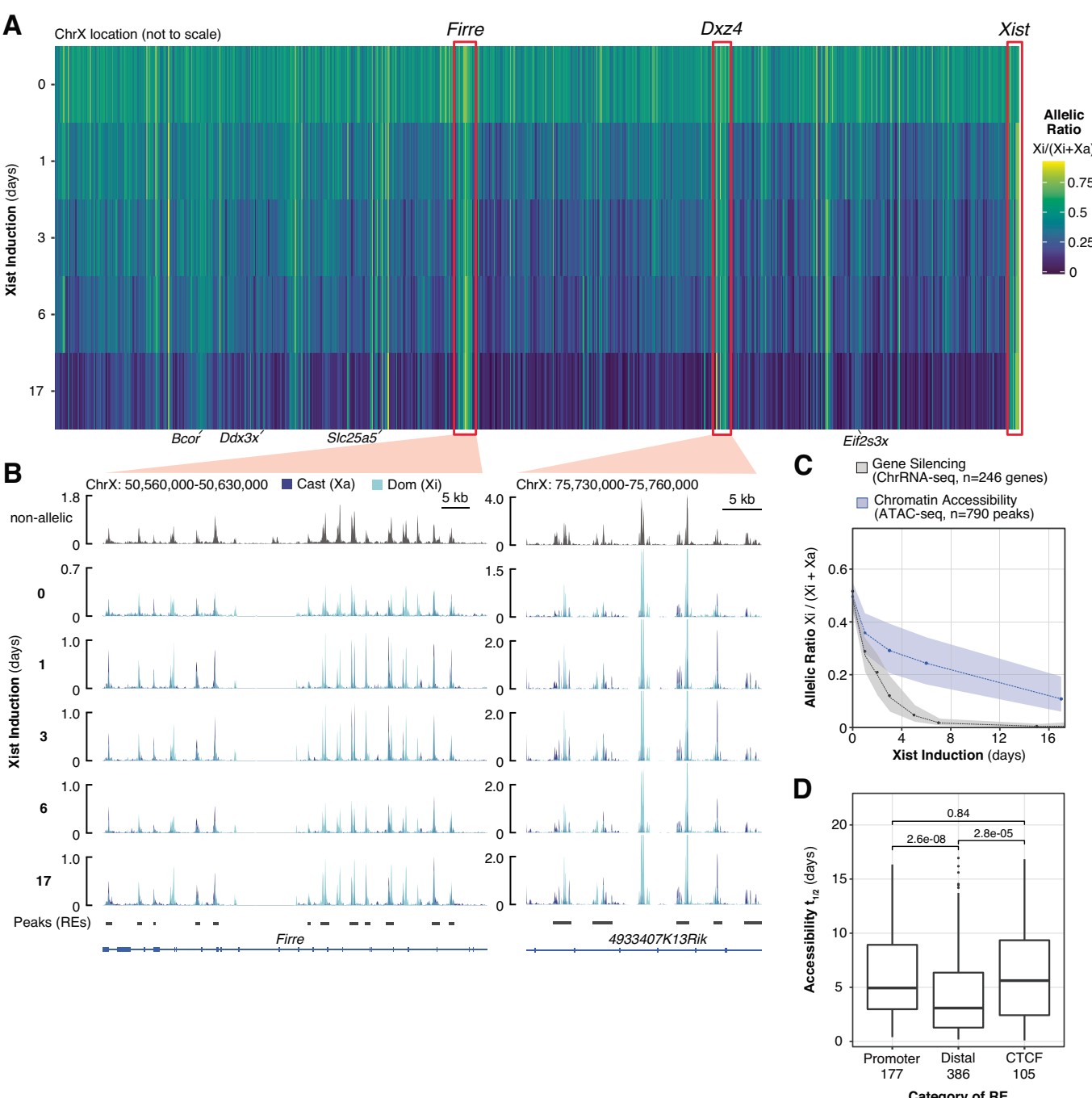

**Figure EV1.  Allele-specific ATAC-seq over a time course of inducible XCI.**

(A) Heatmap of allelic ratios for each X-linked RE in iXist-ChrX_Dom cells over a time course of Xist induction with NPC differentiation. Three loci containing clusters of RE which increase in accessibility on Xi (*Firre*, *Dxz4* and *Xist*) are indicated by red boxes. Locations of escapee gene loci which retain biallelic accessibility *Ddx3x*, *Slc25a5* and *Eif2s3x* are also labelled. *Bicor* is efficiently silenced but contains a cluster of intragenic REs which remain biallelically accessible. Allelic ratios are the average of 2 or 3 replicates for each timepoint. Note that REs on ChrX distal to *Xist* (i.e., 103–165 Mb) are not amenable to allelic analysis due to a recombination event during derivation of the iXist-ChrX_Dom line (Nesterova et al, 2019). (B) ATAC-seq genome tracks of *Firre* and *Dxz4* loci over a time course of Xist induction with NPC differentiation. Tracks are the average of duplicates for each timepoint. (C) Ribbon plot comparing the dynamics of decreasing RE chromatin accessibility with gene silencing dynamics. Dashed lines connect median averages for each timepoint and shaded areas trace interquartile ranges. (D) Boxplots of accessibility halftimes grouped by a categorisation of regulatory elements as promoters (+/-500bp TSS), CTCF sites (CTCF ChIP-seq from GSE144336) or non-CTCF distal elements. Boxes span first to third quartiles with whiskers extending to 1.5 * the interquartile range and a central median line. Significance (*P* values) calculated by Wilcoxon rank-sum test. Numbers of REs in each category are indicated below.

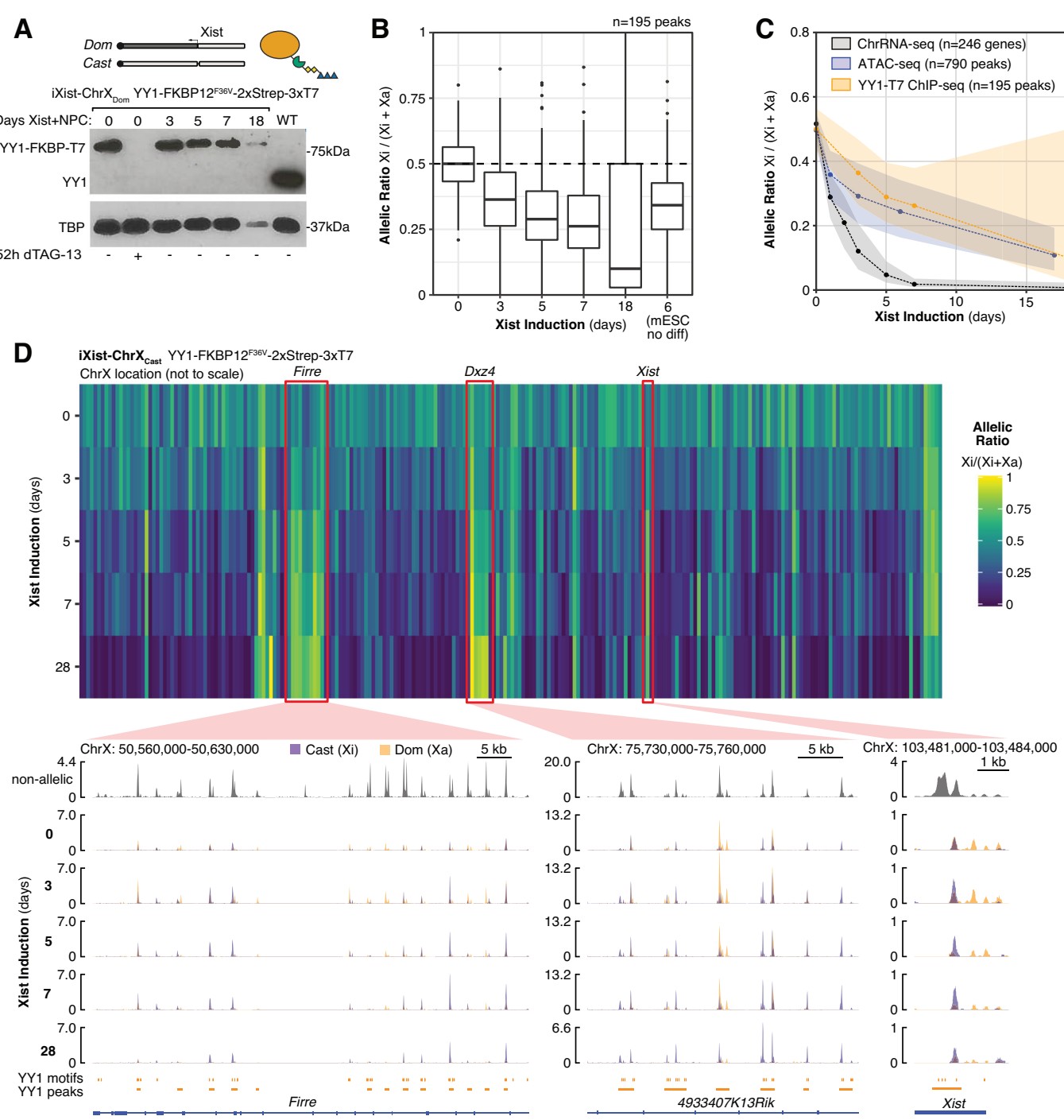

**Figure EV2. YY1-T7 ChIP-seq over a time course of inducible XCI.**

(A) Above: Schematic of the iXist-ChrX$_{Dom}$ YY1-FKBP12$^{F36V}$-2xStrep-3xT7 cell line. Below: Western blot showing expression of the YY1-degron fusion protein in mESCs and over the course of NPC differentiation, as well as acute degradation upon addition of dTAG-13. TBP acts as a loading control. (B) Boxplot of decreasing YY1 binding on Xi, as measured by the allelic ratio in peak regions, over a time course of Xist induction with NPC differentiation (iXist-ChrX$_{Dom}$ background). Boxes span first to third quartiles with a central line indicating the median. Whiskers extend to 1.5 * the interquartile range and outliers outside this range are plotted as separate points. (C) Ribbon plot comparing the dynamics of decreasing YY1 binding with decreasing chromatin accessibility and with gene silencing (iXist-ChrX$_{Dom}$ background). Dashed lines connect median averages for each timepoint and shaded areas trace interquartile ranges. (D) Above: Heatmap of YY1-T7 ChIP-seq allelic ratios for each X-linked YY1-binding peak over a time course of Xist induction with NPC differentiation (iXist-ChrX$_{Cast}$ background). Below: YY1-T7 genome tracks showing three loci important for the 3D organisation of Xi (Firre, Dxz4 and Xist) where Xi-specific YY1 binding increases over the time course of Xist induction.

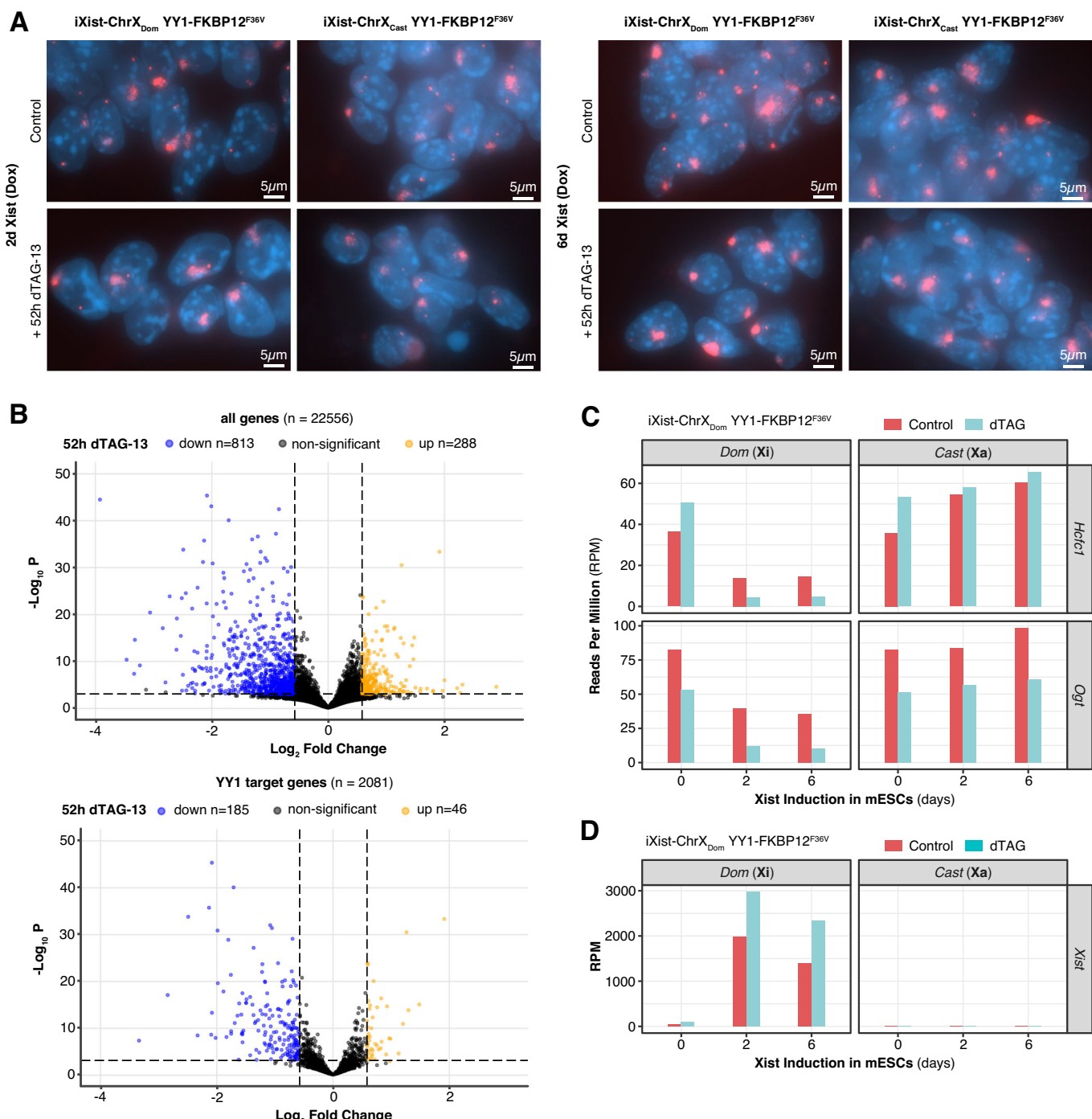

**Figure EV3. Xist RNA localisation and gene expression changes upon YY1 degradation.**

(A) Maximal Z-projection images from Xist RNA-FISH performed on YY1-FKBP12^F36V-2xStrep-3xT7 mESCs after 2 days of Xist induction $+/-$ 52 h dTAG-13 (left) or 6 days of Xist induction $+/-$ 52 h dTAG-13 (right). (B) Enhanced Volcano plot of genome-wide gene expression changes upon 52 h YY1 degradation. All samples (6 dTAG-treated and 6 control) were included in the analysis regardless of Xist induction status. Thresholds to define strongly differentially expressed genes were set at fold change >1.5 and adjusted $P$ value < 0.01 (Wald test with Benjamini–Hochberg correction using DESeq2 (Love et al, 2014)). The lower plot only shows genes defined as direct "YY1 targets" by the presence of a promoter YY1 ChIP-seq peak. (C) Relative allelic expression of two example YY1 target genes, *Hcfc1* and *Ogt*, in ChrRNA-seq experiments performed in the iXist-ChrX_Dom line. (D) Levels of chromatin-associated Xist RNA upon YY1 degradation in ChrRNA-seq experiments performed in the iXist-ChrX_Dom line.

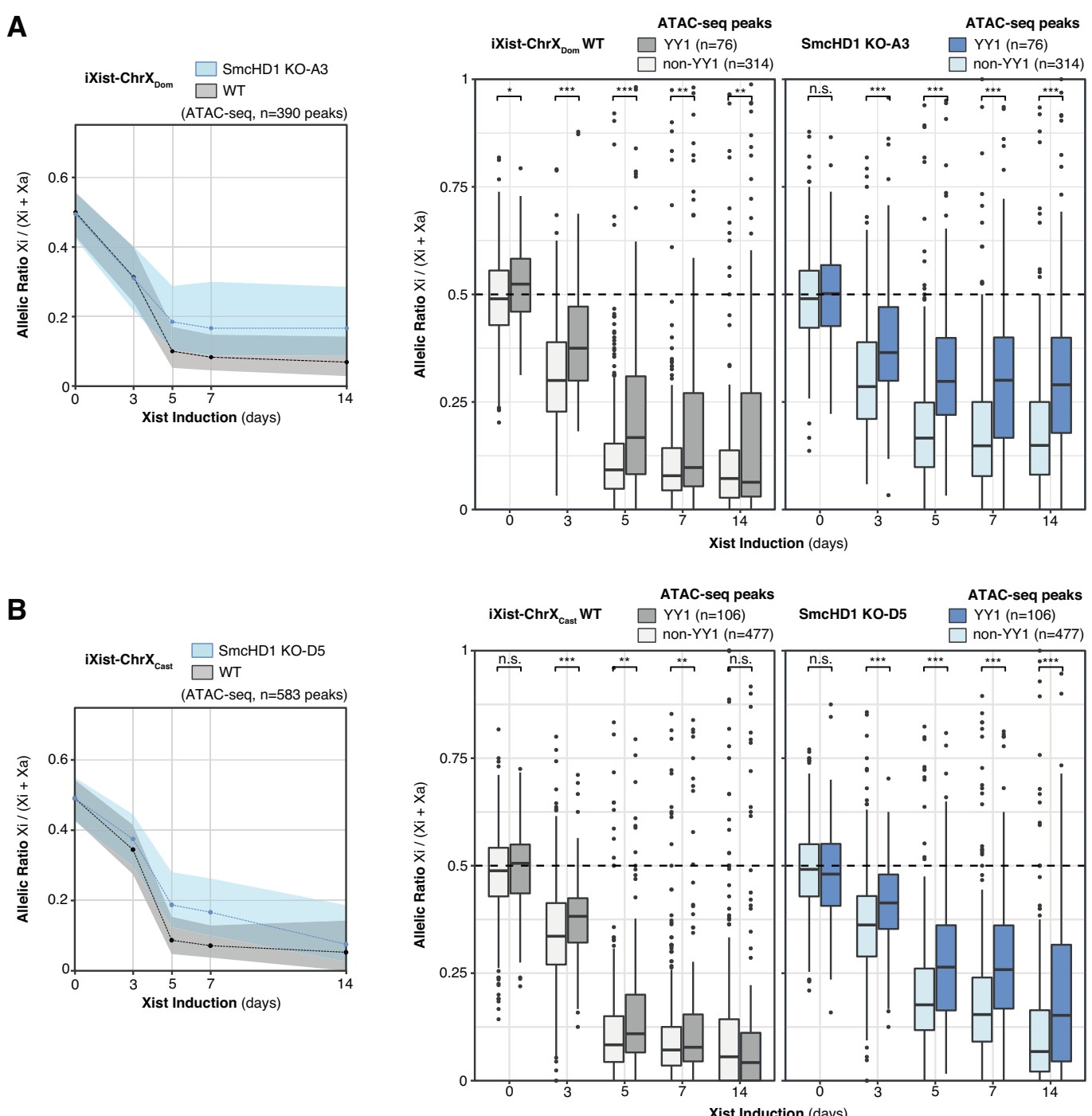

**Figure EV4. Allele-specific ATAC-seq of SmcHD1 KO cell lines upon Xist induction with NPC differentiation.**

(A) Left: Ribbon plot comparing Xi accessibility of ATAC-seq peaks in SmcHD1 KO cell line A3 to wild-type (WT) iXist-ChrX_Dom. Greater residual Xi accessibility in SmcHD1 KO is evident at later timepoints of Xist induction with NPC differentiation, after the time window of SmcHD1 recruitment to Xi in iXist-ChrX cells (between days 3 and 5). Right: Boxplots comparing allelic ratios of YY1 binding versus non-YY1 REs over the ATAC-seq time courses in iXist-ChrX_Dom WT and SmcHD1 KO-A3. Boxes span first to third quartiles with a central line indicating the median. Whiskers extend to 1.5 * the interquartile range and outliers outside this range are plotted as separate points. Significance calculated by Wilcoxon rank-sum test. *, ** and *** indicate P values below, 0.01, 0.001 and 0.0001, respectively. The numbers of YY1-binding and non-YY1 ATAC-seq peaks (REs) which are amenable to allelic analysis are displayed above the plot. (B) As (A) but for SmcHD1 KO cell line D5 derived from the reciprocal iXist-ChrX_Cast line.

