## [Peer Review File · EMBO Reports]

YY1 binding is a gene-intrinsic barrier to Xist-mediated gene silencing

Joseph Bowness, Mafalda Almeida, Tatyana Nesterova, and Neil Brockdorff

Corresponding author(s): Neil Brockdorff (neil.brockdorff@bioch.ox.ac.uk)

Review Timeline:

Transfer from Review Commons:	23rd Feb 24
Editorial Decision:	19th Mar 24
Revision Received:	26th Mar 24
Accepted:	3rd Apr 24

Editor: Esther Schnapp

**Transaction Report: This manuscript was transferred to
EMBO reports following peer review at Review Commons.**

**Review
COMMONS**

Review #1

1. Evidence, reproducibility and clarity:

Evidence, reproducibility and clarity (Required)

****Summary:**** This manuscript uses differentiation of the highly informative inter-specific hybrid mouse ESC to follow features of genes that inactivate slowly. Resistance to silencing is reflected in reduced change in chromatin accessibility and the authors identify YY1 and CTCF as enriched amongst these 'slow' genes. This finding is provocative as these factors have been reported to enrich at both human and mouse escape genes. The authors go on to demonstrate that eviction of YY1 is slowly evicted from the X, and that removal of YY1 increases silencing.

****Minor Comments:****

Overall, the manuscript's conclusions are well supported; however, the brevity of the presentation in some places made it difficult to follow, and in other places seemed a missed opportunity to more fully examine or present their data.

1. Introduction is only 2 paragraphs and half of the last is their new findings. First part of results/discussion is then forced to be very introductory. In addition, some discussion of escapees, even if predominantly human, seems warranted in the introduction. There are multiple studies that have tried to identify features enriched at genes that escape inactivation that could be mentioned.
2. Variation in silencing rates. 'Comparable rankings' cites multiple studies (oddly previous sentence cites only two) - how concurrent are they? Developing this further (perhaps a supplementary table) would inform whether the genes assessed are ones that routinely behave similarly across different studies/lines; and also serve as a resource for future studies.
3. It would be helpful to give insight into informativity of cross - what proportion of ATAC-seq peaks were informative with allelic information (and similarly, what proportion of genes expressed had allelic information?)
4. P5: "can be influenced by Xist RNA via a variety of mechanisms" seems like it this sweeping statement could use expansion, or at least a reference. Authors could also clarify that 'distal elements assigned by linear genomic proximity is their definition of nearest gene.'
5. Figure S1 - there are 6-8 other regions that fail to become monoallelic - what are they?
6. Is there any correlation between silencing speed and expression (as previously

reported)? If yes, then is there also a correlation with YY1 presence - and is this correlation greater than or less than seen on autosomes?

7. Figure 2: Have the authors considered using quartiles rather than an arbitrary division into depleted and persistent? B. Could simplify secondary labels to solely YY1 and CTCF. D & F do not print in black and white. Overall the mESC versus NPC can be confusing, perhaps mESC (no diff) would be helpful?

8. The numbers appear to suggest YY1 is generally enriched on X, but not at promoters?? Is this true? For Figure 2f, it might be helpful to show autosomal genes - are Fast depleted or Slow enriched for YY1 relative to autosomes? More generally, is YY1 binding on the X lost more slowly than YY1 binding on autosomes, or is the slow loss a feature of YY1. While I agree YY1 could have direct up or down-regulatory roles, Figure S3 could also be reflecting a secondary impact.

9. Figure 3, 4 and supplementary - the chromosome cartoon introduces the LOH in iXist, but this needs to be described in text. Describing the reciprocal as a biological replicate seems challenging given this LOH.

10. Could a panel of TFs be used rather than OCT4 which has its own unique properties to emphasize that YY1 is unique?

11. Figure 4 - by examining 'late' genes, a change in allelic ratio is observed, but what about escape genes (e.g. Kdm5c, Kdm6a)? Do they now become silent? It would be helpful to have all this data as a supplementary table so people could query their 'favorite' gene.

12. It seems surprising that loss of YY1 has no demonstrative impact on the Xa. Figure S3B suggests that over 1000 genes are significantly impacted - primarily down regulated. How many of those are X-linked? Perhaps they could be colored differently?

13. Since XIST+/undifferentiated cells retain YY1, is YY1 binding sensitive to DNase? Indeed, are X chromosome bound sites in islands that become methylated? Figure S4 shows YY1-targetted X genes in SMCHD1 knockout; can CTCF targets also be shown? While identified in Figure 2, CTCF was not examined the way YY1 was, although it has also been identified in somatic studies of genes that escape X inactivation.

14. When the authors use cf. do they simply mean see also, or as wikipedia suggests: "the cited source supports a different claim (proposition) than the one just made, that it is worthwhile to compare the two claims and assess the difference". Perhaps it would be worth spelling out to clarify for the audience.

2. Significance:

Significance (Required)

General assessment: An important question in human biology is how much the sex chromosome contributes to sex differences in disease frequency. Genes that escape X inactivation in humans seem to have considerable impact on gene expression genome-wide. While there are not as many genes in mouse that escape inactivation, the use of the mESC cell differentiation approach allows detailed assessment of the timing of silencing during inactivation. The authors utilize an inter-specific cross and it would be interesting to know the limitations of such a system (in terms of informative DHS/genes that are informative).

Advance: As the authors note, there are multiple studies of similar systems that have revealed differences in the speeds of silencing of genes. However, this is the first study to my knowledge that has then tried to assess timing with gene-specific factors. There are multiple studies in humans comparing escape and subject genes for TFs, but lacking the developmental timing that this study incorporates.

Audience: While generally applicable to a basic research audience interested in gene regulation, the applicability to human genes that escape inactivation may interest cancer researchers or clinical audiences interested in sex differences.

3. How much time do you estimate the authors will need to complete the suggested revisions:

Estimated time to Complete Revisions (Required)

(Decision Recommendation)

Between 1 and 3 months

Yes

Review #2

1. Evidence, reproducibility and clarity:

Evidence, reproducibility and clarity (Required)

The authors studied the molecular basis of a variation in the rate of individual gene silencing on the X undergoing inactivation. They took advantage of ATAC-seq to observe the kinetics of chromatin accessibility along the inactive X upon induction of Xist expression in mESCs. They demonstrated a clear correspondence between the decrease in chromatin accessibility and the silencing of nearby genes. Furthermore, they found that persistently accessible regulatory elements and slow-silencing were associated with binding of YY1. YY1 tended to associate longer with genes that required more time to be silenced than those that became silenced fast on the inactive X during XCI. The acute loss of YY1 facilitated silencing of slow genes in a shorter period. They suggest that whether or not the transcription factors stay associated longer is another factor that impacts the variation in the rate of gene silencing on the Xi.

2. Significance:

Significance (Required)

It has been suggested that the rate of gene silencing during XCI varies depending on the distance of individual genes from the Xist locus or the entry site of Xist RNA on the X, as well as their initial expression levels before silencing. This study provides another perspective on this issue. The persistent association of transcription factors during XCI affects the rate of gene silencing. Although the issues addressed here might draw attention from only the limited fields of specialists, their findings advance our understanding of how the efficiency of silencing is controlled during the process of XCI. The experimental data essentially support their conclusion, and the manuscript was easy to follow. However, I still have some comments, which I would like the authors to consider before further consideration.

****Major concerns****

Based on the results shown in Figure 3E and F, the authors concluded that YY1 was

more resistant than other TFs against the eviction from the X upon Xist induction. I am not still convinced with this. YY1 binds DNA via the zinc finger domain, while Oct4 binds DNA via the homeodomain. The difference in the binding module between them might affect their dissociation or the response to Xist RNA-mediated chromatin changes. In addition, given that YY1 has been reported to bind RNA, including Xist, as well, Oct4 might not be a good TF to compare.

I don't think that Kinetics of YY1 eviction upon Xist induction in SmcHD1 KO cells during NSC differentiation fit the phenotype of Smchd1 mutant cells. Although their previous study by Bowness et al (2022) showed that Smchd1-KO cells fail to establish complete silencing of SmcHD1-dependnet genes, their silencing still reached rather appreciable levels according to Figure 6 of Bowness et al (2022). This is, in fact, consistent with the idea that XCI initially takes place in the mutant embryos, at least to an extent that does not compromise early postimplantation development. On the other hand, a significant portion of YY1 appears to remain associated with the target genes on both active and inactive X (Figure S4), which I think suggests that the presence of YY1 is compatible with silencing of SmdHD1-dependent genes. This is contradictory to the proposed role of YY1 that sustains the expression of X-linked genes in this context.

3. How much time do you estimate the authors will need to complete the suggested revisions:

Estimated time to Complete Revisions (Required)

(Decision Recommendation)

Between 1 and 3 months

Yes

Review #3

1. Evidence, reproducibility and clarity:

Evidence, reproducibility and clarity (Required)

In this manuscript, Bowness and colleagues describe the interesting finding that the transcription factor YY1 is associated with slow silencing genes in induced X-Chromosome Inactivation (XCI). The authors have conducted a comprehensive characterization of X-linked gene silencing and the loss of chromatin accessibility of regulatory elements in induced XCI in ESCs and during NPC differentiation. X-linked gene silencing was classified into four categories, ranging from fast-silenced genes to genes that escape silencing. Motif enrichment analysis of regulatory elements associated with slowly silenced genes identified YY1 as the transcription factor most significantly enriched. The separation of YY1-target and non-target genes confirmed that most genes bound by YY1 indeed exhibit slower silencing kinetics. A comparison of the binding kinetics of YY1 to another transcription factor, OCT4, during XCI revealed that YY1 is evicted more slowly compared to OCT4 on the inactive X, suggesting that slower eviction is a unique property of YY1. Conditional knock-outs of YY1 using protein degradation during induced XCI in mESCs demonstrated that the loss of YY1 at target genes enhances silencing. This supports the hypothesis that YY1 serves as a crucial barrier for slow-silenced genes during XCI. Finally, the authors propose a hypothesis regarding the mechanism of YY1 eviction, suggesting a potential connection to the role of SmcHD1 during XCI.

The authors provide an in-depth analysis of the role of YY1 in gene silencing kinetics during induced XCI and believe this manuscript should be published if our comments are addressed.

****Major comment:****

Based on the allelic ratio in figure 3C only minor loss of YY1 binding occurs in induced XCI in mESCs on the Xi, while silencing is established properly as shown in figure 4C (left panel, red boxplots). This suggests that YY1 eviction is not necessarily required for these genes to be effectively silenced. Could the authors explain this discrepancy in the data regarding their manuscript conclusions? It seems this is true for XCI happening during differentiation towards NPCs, but not if cells are stuck in the pluripotency stage?

****Minor comments:****

In the abstract lines 7-8, the authors state that the experiments were performed in mouse embryonic stem cell lines, but much of the data shown is acquired in NPC differentiations. Please adjust abstract.

The last sentence of the abstract states that YY1 acts as a barrier to silencing but as stated in my major comment, that does seem to be the case in ESC differentiation towards NPCs, but not in ESCs themselves. Please tone down this sentence. Moreover, we do not fully understand where the 'is removed only at late stages' comes from? Is this because of the *Smchd1* link? We find this link quite weak with the data presented. We would tone down that last abstract sentence.

Several comparisons to human XCI have been made in the article. We do agree that there are similarities between mouse and human XCI. However, there is insufficient data that substantiates that these genes are regulated in a similar manner in humans. We believe the comparisons should be removed altogether or attenuated.

At bottom of page 4, the authors say that for any given gene, the allelic ration of accessibility at its promoter decreased more slowly than it silenced and then write Fig 1B. They probably mean S1C? Since 1B only shows 4 genes.

Figure 1B shows the average allelic ratio of multiple clones for genes representing different silencing speeds. Each data point is the average of multiple clones for these representative genes, could the authors show the individual data points or the standard deviation?

Figure 1B. Loss of promoter accessibility lags behind loss of chromatin-associated RNA expression for these 4 genes. What about distal REs? Do the allelic ratios for the

distal REs more closely follow chromatin-associated RNA expression? Could the authors show this in a supplemental figure?

Figure 1B: gene silencing trajectory is depicted left while the legend says right. Same for promoter accessibility.

Figure S1A shows only part of the X chromosome. The area downstream of Xist is missing. Is this because the iXist-ChrXDom cell line is missing allelic resolution as shown in figure S2A? Could the authors explain in the figure legend that part of the X-Chromosome is missing?

Figure 2C shows that 94 TSSs bear a YY1 peak, yet Fig 2F shows 62 are targets of YY1. Is this because the rest are not properly silenced or are escapees?

Moreover, YY1 has ~4-fold more peaks on the X chromosome on distal elements compared to promoters. Yet figure 2F exclusively shows the proportion of YY1 binding sites on TSSs. Would distal REs show similar proportions for the silencing categories? Could the authors show the differences in a Supplemental figure?

The authors switch between different model systems in the figures, which makes quite confusing which type of XCI is being discussed. We would like to see clearly stated above all panels which cell culture condition is being studied (mESCs or NPCs).

In Figure 3E and 3F the authors look at the binding retention of OCT4 during XCI in ESCs. However, it is not clear why the authors choose OCT4. Could the authors explain why specifically OCT4 was chosen for these analyses?

What is the expression level of YY1 in NPCs compared to mESCs? In Supplemental S2A, it seems that YY1 protein levels decrease over time during NPC differentiation. Is part of the increased eviction a result of lower protein levels of YY1? Probably not since you calculate ratios between Xi and Xa. Can you please comment on this?

On page 7 the authors state that degrading YY1 does not affect Xist spreading and/or localisation. Indeed, it has been previously shown by other groups that YY1 is required for Xist localisation during XCI. Could the authors elaborate further on the why their cells behave differently compared to the Jeon 2011 paper?

In figure 4G and figure S3D elevated levels of Xist are observed in the dTAG conditions. As the authors point out, this could then result in accelerated silencing of the X seen upon YY1 loss. Are these elevated Xist levels that result in enhanced silencing in figure 4 relevant for the kinetics of silencing? Moreover, YY1 could act

as transcriptional regulator of those genes in the X and by removing YY1, one would expect decreased transcription, which would be read as accelerated silencing. The authors could see whether the genes that show accelerated silencing are regulated by YY1 in ESCs (+ dTAG, - Dox).

Can the authors explain why they decided to put the Smchd1 part after the conclusion? Before the conclusion would have been better? The probable link between YY1 and SmcHD1 is definitely something important to investigate.

In the paper the authors cite Bowness et al., 2022. In it, Figure 5F studies silencing times with respect to silencing dependency on SmcHD1. What is the overlap between SmcHD1 target genes and YY1 target genes? This would provide more data about the correlation between YY1 and SmcHD1.

The authors hypothesise that SmcHD1 might play a role in the eviction of YY1 in NPC differentiation. The current data shows impaired silencing of slow silencing genes and YY1-dependent genes in the SmcHD1 knock-out. However, it doesn't show SmcHD1 is required for YY1 eviction. Could the authors provide direct evidence for their hypothesis by performing NPC differentiation in wild type and SmcHD1 knock-out cells and investigate YY1 binding using ChIP-seq?

In figure S4A and S4B no significance is indicated among the different conditions across the different differentiation days. Could the authors add this?

Finally, we would like the authors to elaborate in the conclusion about the order of events. As they correctly state at the top of page 5 (and we agree), delayed loss of promoter accessibility compared to gene silencing does not automatically mean that it is downstream of gene silencing. Can you elaborate on this? Also, in light of Fig S2C where loss of YY1 binding seems to happen after gene silencing.

2. Significance:

Significance (Required)

This manuscript highlights a novel role for YY1 in XCI. The manuscript provides an analysis of the correlation and causation of YY1 in gene silencing during XCI. There is a clear correlation between YY1 and delayed silencing of genes on the Xi. To our knowledge, this is the first time such an analysis has been performed for YY1. It advances our conceptual and mechanistic understanding of gene silencing kinetics and what the factors involved in it are. We believe it is an important contribution to the XCI field and will be of great value to the XCI community.

Strength:

This study presents a comprehensive and in-depth characterization of X-linked gene silencing during XCI.

Two different types of inducible XCI are studied and compared (ESCs vs differentiation towards NPCs), which we are grateful for.

Systematic and stepwise analysis of the data is very strong.

Many data points have been collected which provide stronger conclusions.

Weakness:

Some sentences in the abstract should be toned down.

YY1 eviction on the inactive X doesn't seem crucial to establish X-linked gene silencing in mESCs.

The mechanistic approach at the end of the manuscript with relation to SmcHD1 could be studied further.

This paper will be suited for a specialised audience in XCI and transcription factor control of gene expression, i.e. basic research.

Field of expertise: XCI, epigenetics, Xist, gene silencing, X chromosome biology.

3. How much time do you estimate the authors will need to complete the suggested revisions:

Estimated time to Complete Revisions (Required)

(Decision Recommendation)

Between 3 and 6 months

Yes

Full Revision

Manuscript number: RC-2023-02271

Corresponding author(s): Neil, Brockdorff

[Please use this template only if the submitted manuscript should be considered by the affiliate journal as a full revision in response to the points raised by the reviewers.

If you wish to submit a preliminary revision with a revision plan, please use our "Revision Plan" template. It is important to use the appropriate template to clearly inform the editors of your intentions.]

1. General Statements [optional]

We would like to thank all three reviewers for their careful and comprehensive reviews of our manuscript. We have taken on board all the comments and have made appropriate changes to improve the manuscript. The more substantive changes are to the structuring of the text in Introduction section, and to improving the clarity of Figure 2 after reviewers' comments (we have added extra panels to A, F and G). Other minor changes are individually signposted in each paragraph of the point-by-point response attached below.

We performed a number of pieces of additional analysis to address reviewer comments. To be as transparent as possible we make these and all other data analyses available in the form of .html files exported by Rmarkdown, hosted at <https://joebowness.github.io/YY1-XCI-analysis/>.

This section is mandatory. Please insert a point-by-point reply describing the revisions that were already carried out and included in the transferred manuscript.

Reviewer #1 (Evidence, reproducibility and clarity (Required)):

Summary: This manuscript uses differentiation of the highly informative inter-specific hybrid mouse ESC to follow features of genes that inactivate slowly. Resistance to silencing is reflected in reduced change in chromatin accessibility and the authors identify YY1 and CTCF as enriched amongst these 'slow' genes. This finding is provocative as these factors have been reported to enrich at both human and mouse escape genes. The authors go on to demonstrate that eviction of YY1 is slowly evicted from the X, and that removal of YY1 increases silencing.

Minor Comments:

Overall, the manuscript's conclusions are well supported; however, the brevity of the presentation in some places made it difficult to follow, and in other places seemed a missed opportunity to more fully examine or present their data.

Full Revision

1. Introduction is only 2 paragraphs and half of the last is their new findings. First part of results/discussion is then forced to be very introductory. In addition, some discussion of escapees, even if predominantly human, seems warranted in the introduction. There are multiple studies that have tried to identify features enriched at genes that escape inactivation that could be mentioned.

We have now written the introduction as 3 paragraphs instead of 2. In doing this, we have moved the sentence introducing chromatin accessibility from the results section to the introduction. Additionally, we now discuss the studies that focus on escapees (in mouse XCI) in the second introduction paragraph.

2. Variation in silencing rates. 'Comparable rankings' cites multiple studies (oddly previous sentence cites only two) - how concurrent are they? Developing this further (perhaps a supplementary table) would inform whether the genes assessed are ones that routinely behave similarly across different studies/lines; and also serve as a resource for future studies.

To avoid double-citing, we have made this one sentence and have cited at the end of the sentence 7 studies which describe gene-by-gene variability in rates of silencing. The majority of these studies include comparisons of their categories of fast and slow-silencing gene with previous classifications, and they all conclude that there is substantial concurrence. Some examples:

- Marks et al, 2015, Table S3,
- Loda et al, 2017, Figure 5,
- Barros de Andrade E Sousa et al. 2019, Figure 2
- Pacini et al. 2021, Figure 6e,i

We believe this is sufficient evidence for our claim that these studies report “comparable categories” (“ranking” changed to “categories” as not all studies strictly rank). A comprehensive gene-by-gene comparison table would likely serve only to highlight differences due the various silencing assays/model systems/classification approaches used in the studies. If required, however, we would be willing to include a supplemental table which collates where gene silencing categories are discussed in each publication, and links to any supplemental files which provide full lists of X-linked genes.

3. It would be helpful to give insight into informativity of cross - what proportion of ATAC-seq peaks were informative with allelic information (and similarly, what proportion of genes expressed had allelic information?)

Of the 2042 consensus ATAC-seq peaks we defined on ChrX via aggregating macs2 peaks over all time course samples, n = 821 passed our initial criteria for allelic analysis in the iXist-ChrX-Dom model line (ie they are proximal to the *Xist* locus in ChrX 0-103Mb, overlap SNPs, and contain sufficient allelic reads). A small number of peaks were additionally filtered out during fitting of the exponential decay model, leaving a final ATAC-seq peak set of n = 790 elements (38.6%) which we focus on in this study. We have added this information to the text (first Results paragraph).

Our collections of ChrX genes amenable to allelic analysis were not redefined for this study. We used lists of genes defined in our previous ChrRNA-seq study (10.1016/j.celrep.2022.110830). In general, allelic analysis of gene expression is not as limited by the frequency of SNPs, because the

Full Revision

sequence length of transcripts (including introns, which are a significant fraction of the reads in ChrRNA-seq data) is much greater than for ATAC-seq peaks. Only a few very lowly expressed genes are not amenable to allelic ChrRNA-seq analysis.

4. P5: "can be influenced by Xist RNA via a variety of mechanisms" seems like it this sweeping statement could use expansion, or at least a reference. Authors could also clarify that 'distal elements assigned by linear genomic proximity is their definition of nearest gene.

The statement that "both [chromatin accessibility and gene expression] can be influenced by Xist RNA via a variety of mechanisms" is intentionally broad to support a negative argument that we do not wish to mechanistically over-interpret the observation that Xi chromatin accessibility loss occurs slower than gene silencing. Nonetheless, we have added two references to studies which report mechanisms for how Xist may influence chromatin accessibility; via recruiting PRC1 (Pintacuda et al 2017) or antagonising BRG1 (Jegu et al 2019). That multiple molecular pathways simultaneously contribute towards the effect of Xist RNA on gene silencing is well established in the field (see reviews such as Brockdorff et al 2020, Boeren et al 2021, Loda et al 2022).

We have clarified in the text that our definition of "distal" is all REs which do not overlap with promoter regions (TSS \pm 500bp). We have also made it clearer that our definition of "nearest" gene refers to linear genomic proximity in both the Results and Methods sections.

5. Figure S1 - there are 6-8 other regions that fail to become monoallelic - what are they?

The regions which stand out most by the colour scheme of the heatmap in Figure S1 are those where accessibility *increases* on Xi, most notably the loci of *Firre*, *Dxz4* and *Xist*, which are known to have unique features related to the 3D superstructure of the inactive X chromosome. A few other regions which do not become monoallelic harbour classic "escapee" genes. We have now labelled the locations of escapees *Ddx3x*, *Slc25a5* and *Eif2s3x* in FigS1.

The other regions noticeable in the heatmap have no obvious features which explain why they fail to become monoallelic. We have highlighted a region containing intragenic peaks within *Bcor* (a gene which is silenced in iXist-ChrX mESCs), but many other regions are not in the vicinity of genes. Some of the persistently Xi-accessible peaks within these regions contain strong YY1 or CTCF sites, although many others do not.

It is also possible that some Xi-accessible peaks are artefacts of mismatches between the *Castaneous* or *Domesticus/129Sv* strain SNP databases and ground truth iXist-ChrX genome sequence. The number of these cases are small, and if a misannotated SNP is the only SNP present in a single peak, the peak is discarded by our allelic filtering criteria as it will appear monoallelic in uninduced mESCs.

6. Is there any correlation between silencing speed and expression (as previously reported)? If yes, then is there also a correlation with YY1 presence - and is this correlation greater than or less than seen on autosomes?

Full Revision

The data we present here pertaining to gene silencing kinetics is reused from our previous study. In that work we did indeed observe a significant association between silencing rate and initial gene expression levels (10.1016/j.celrep.2022.110830, Supplemental Information Figure S5F), which has also been reported by multiple groups previously.

To correlate YY1 binding with gene expression levels, we calculated transcripts per million (TPM) for all genes from our genome-wide mRNA-seq data of uninduced iXist-ChrX-Dom cells (GSE185869). It is indeed true that, on average, X-linked genes classified as “direct” YY1-targets in our analysis have higher levels of initial expression (median TPM 70.8, n=64) compared to non-target genes (median TPM 30.7, n=346). Autosomal YY1 targets are also relatively higher expressed (median TPM 29.6, n=1882) than non-YY1 genes (median TPM 8.0, n=9983). Within the list of YY1-targets, there is no additional correlation between quantitative levels of YY1 ChIP enrichment (calculated in this study using BAMscale (Pongor et al, 2020)) and gene expression ($R=-0.05$, Spearman correlation).

Therefore, we appreciate that this correlation between YY1-binding and gene expression levels may be a covariate in the correlation we report in this study between YY1-target genes and slow-silencing. This does not invalidate a potential functional role for YY1 in impeding silencing, as it could affect both variables via common or distinct mechanisms. Nevertheless, in an attempt to account for initial expression level as a covariate, we compared the silencing halftimes of YY1-targets versus non-targets within genes grouped by similar expression levels (low, medium and high-expressed genes). YY1-targets have slower halftimes in each comparison, and this difference is highly significant ($p=1.9e-05$, Wilcoxon test) for the “medium-expressed” gene group. This implies that YY1 contributes towards slower gene silencing kinetics independently of initial gene expression levels. We have added this panel to Fig2 with an associated sentence in the Results section.

These new analyses are also appended to the documentation of the R scripts used to generate the main figures in this study (Figure2_YY1association.Rmd), which will all be published to Github.

It is also important to note that this analysis approach is complicated by the methodology we use to classify YY1 target genes. In this study, we define YY1 targets based on the presence of ChIP-seq peaks overlapping the gene promoters, which is reasonable and widely accepted practice when defining targets of transcription factors. However, as briefly discussed in the Methods, in YY1 ChIP-seq data samples with very high signal:noise (eg Fig3), minor peaks of YY1 enrichment can be detected at almost every active promoter. As enrichment at these peaks is typically much less than at peaks with occurrences of the YY1 consensus DNA motif, we hypothesise that these small peaks result from secondary YY1 cofactors enriched at promoters (eg P300, BAF, Mediator) rather than direct sites of binding to DNA/chromatin. Therefore, for annotating genes as “direct” YY1 targets, we chose to use the YY1 peak set defined from lower signal:noise ChIP-seq data in iXist-ChrX produced with the endogenous YY1 Ab. Nevertheless, this behaviour is likely to confound any analysis correlating YY1 ChIP binding with gene expression.

7. Figure 2: Have the authors considered using quartiles rather than an arbitrary division into depleted and persistent?

Full Revision

We primarily chose this binary classification of REs as either Xi-“persistent” or Xi-“depleted” to maximise the numbers of sequences that could be used in each group as input for the HOMER motif enrichment software.

It is also not trivial to separate REs into quartiles because our “Xi-persistent” classification includes peaks defined as “biallelically accessible in NPCs”, as well as peaks with slow accessibility halftimes. This is explained in both the Results and Methods but we now have edited Fig2A to make it clearer. Instead of quartiles, we have performed an analysis which keeps “biallelically accessible REs” as a separate category and subdivides the remaining peaks into three groups by halftimes (slow, intermediate and fast accessibility loss). The same trends are evident with this four-category approach as with the two-category approach.

Importantly, our follow-up analyses which confirm the association between YY1 binding and slow Xi accessibility loss (Fig2E) and slow silencing (Fig 2F-H) are independent from categorisations of REs which rely on arbitrary thresholds.

B. Could simplify secondary labels to solely YY1 and CTCF. D & F do not print in black and white. Overall the mESC versus NPC can be confusing, perhaps mESC (no diff) would be helpful?

We have simplified the secondary labels in Fig2B and modified the colour scheme of Fig2D and Fig2F as suggested. “mESC” is now modified to “mESC no diff” in Fig2H, FigS2B, Fig3C and Fig3E to reduce the potential for confusion.

8. The numbers appear to suggest YY1 is generally enriched on X, but not at promoters?? Is this true?

The explanation for this is that clear peaks of YY1 ChIP are found at young LINE1 elements in iXist-ChrX mESCs (specifically over L1Md_T subfamilies). These elements are highly enriched (>2-fold) on the mouse X chromosome compared to autosomes (Waterston 2002), and the majority are not promoter-associated. We chose not to include a discussion of YY1 enrichment at repetitive LINE1 elements in this study primarily because of a) issues related to multiple-mapping reads, such as difficulties distinguishing ChrX vs autosomal reads, and b) the absence of strain-specific SNPs within annotated ChrX L1Md_Ts means that none of these elements are amenable to allelic analysis so we cannot compare Xi versus Xa. However, these LINE1 peaks are a significant fraction (262/521) of the numbers of YY1 ChIP-seq peaks in Fig2C.

For Figure 2f, it might be helpful to show autosomal genes - are Fast depleted or Slow enriched for YY1 relative to autosomes?

We have calculated these numbers as part of the analysis of gene expression on ChrX and autosomes above. Overall, the fraction of genes defined as YY1-targets is the same on ChrX as on autosomes (~0.16). Accordingly, fast-silencing genes are depleted for YY1 compared to autosomes, whereas slow-silencing genes are enriched for YY1 compared to autosomes. Fig2F is now redesigned to include the total numbers of YY1-target genes on ChrX and autosomes.

Full Revision

More generally, is YY1 binding on the X lost more slowly than YY1 binding on autosomes, or is the slow loss a feature of YY1. While I agree YY1 could have direct up or down-regulatory roles, Figure S3 could also be reflecting a secondary impact.

We agree that many of the differentially regulated genes after 52 hours of YY1 degradation could be secondary effects and have added a sentence on this to the relevant paragraph in the text.

9. Figure 3, 4 and supplementary - the chromosome cartoon introduces the LOH in iXist, but this needs to be described in text. Describing the reciprocal as a biological replicate seems challenging given this LOH.

It is true that the reciprocal lines iXist-ChrX-Dom and iXist-ChrX-Cast are not true biological replicates, and we try to avoid referring to them as such. Writing this in the legend of Fig3 was an error which we have corrected. We have now also mentioned the recombination event in the iXist-ChrX-Dom cell line at the point where data from this line is first discussed (paragraph 1 of Results section).

For the latter parts this work (Figs 3 and 4), we made the conscious decision to proceed with two YY1-FKP12^{F36V} cell lines from different reciprocal iXist-ChrX backgrounds (aF1 in iXist-ChrX-Dom, cC3 in iXist-ChrX-Cast), rather than “biological replicate” clones from either iXist-ChrX-Dom or iXist-ChrX-Cast. Our reasoning was to control against potential confounding effects of strain background on our experiments related to the role of YY1. Although there were some minor differences between the clones, aF1 and cC3 demonstrated essentially equivalent phenotypes in all analyses we performed.

10. Could a panel of TFs be used rather than OCT4 which has its own unique properties to emphasize that YY1 is unique?

This would indeed be worthwhile, and we did consider attempting to perform ChIP-seq for additional TFs other than OCT4 in order to collect more points of comparison for the slow rate of loss of YY1 binding to Xi. However, it is admittedly hard to identify appropriate candidate TFs in mESCs which a) have similar numbers of discrete peaks of binding in promoters and distal elements on ChrX and b) it is possible to reliably perform ChIP-seq for at sufficiently high signal:noise to allow for quantitative allelic analysis.

We have changed the text to acknowledge that our comparison only to OCT4 limits the scope of the statements we can make about unique properties of YY1 binding.

11. Figure 4 - by examining 'late' genes, a change in allelic ratio is observed, but what about escape genes (e.g. Kdm5c, Kdm6a)? Do they now become silent? It would be helpful to have all this data as a supplementary table so people could query their 'favourite' gene.

YY1 degradation experiments performed for Figure 4 were performed on mESCs without cellular differentiation (YY1-ablated cells do not survive in our mESC to NPC differentiation protocol). In undifferentiated mESCs, silencing of the inactive X does not reach completion, and in fact all X-linked genes are residually expressed at a higher level than in equivalent timepoints of Xist induction with NPC differentiation (see Figure 4D, Bowness et al 2022). We write in the text “slow-silencing

Full Revision

genes are residually expressed from Xi” because genes of this category account for the majority of expression under these conditions, and indeed almost all slow genes would all be classed as “escape genes” in this setting by a conventional definition of >10% residual expression from Xi (see also Figure 4D, Bowness et al 2022). Our analysis in Fig4D (of this study) includes all genes, and we share processed .txt files of allelic ratio and allelic fold changes in GEO, so querying the behaviour of a favourite gene would be easy (GSE240680).

Incidentally, when we do perform NPC differentiation of iXist-ChrX NPC, at late stages very few genes show any expression from Xi (*Ddx3x*, *Slc25a5*, *Eif2s3x* and *Kdm5c* clearly escape, but even *Kdm6a* is entirely silenced). Unfortunately, with such a small number of “super” escapees it is hard to make any general conclusions, so in this study we can only make inferences about escape via the transitive property that many “slow-silencing” genes are facultative escapees in other settings without induced Xist overexpression. We now write about this consideration in the introduction and final paragraph of the main text.

12. It seems surprising that loss of YY1 has no demonstrative impact on the Xa. Figure S3B suggests that over 1000 genes are significantly impacted - primarily down regulated. How many of those are X-linked? Perhaps they could be colored differently?

For the broad-brush differential expression testing in FigS3B, we use all the ChrRNA-seq samples (6 x untreated, 6 x dTAG) as “pseudo-replicates”, disregarding any confounding effects related to induced Xist-silencing as effecting untreated and dTAG sample groups equivalently. We did specifically investigate the behaviour of X-linked genes in this volcano plot, however only a very small number of genes were differentially expressed (n=22 X-linked genes appeared significantly downregulated compared to n=4 genes upregulated). This can be seen in our analysis records uploaded to Github.

Additionally, there is actually a minor effect of YY1 loss on expression of YY1-target genes on Xa. This can be seen in Fig4F, where the median lines of YY1-target boxes lie below the horizontal line of 0-fold change.

13. Since XIST+/undifferentiated cells retain YY1, is YY1 binding sensitive to DNAm? Indeed, are X chromosome bound sites in islands that become methylated? Figure S4 shows YY1-targetted X genes in SMCHD1 knockout; can CTCF targets also be shown? While identified in Figure 2, CTCF was not examined the way YY1 was, although it has also been identified in somatic studies of genes that escape X inactivation.

Binding of YY1 is indeed sensitive to DNA methylation; specifically it is reported to be blocked by CpG methylation (see refs (Kim *et al*, 2003; Makhlouf *et al*, 2014; Fang *et al*, 2019). Thus, crosstalk with the DNA methylation pathways, which deposit *de novo* CpG island methylation as a late event of XCI (Lock 1987, Gendrel 2012), did appeal to us as a potential mechanism of YY1 “eviction”. However, preliminary analysis we performed to investigate this revealed limited overlap between YY1 binding sites and *de novo* methylated CpG islands in the iXist-ChrX model cell line.

Full Revision

FigS4 presents ATAC-seq data from two iXist-ChrX SmcHD1 KO clonal cell lines, comparing the accessibility loss kinetics between YY1-binding and non-YY1 REs in these cells.

Although FigS4 in this paper does not show genes, we have previously published ChrRNA-seq data from these SmcHD1 KO lines over a similar Xist induction + NPC differentiation time course (Figure 6, Bowness et al, 2022). A reanalysis of this ChrRNA-seq data by YY1-target vs non-target genes shows a similar trend to the accessibility data, although this is expected from the strong overlap of both “YY1-target” and “SmcHD1-dependent” genes with slow-silencing genes in our model.

With respect to CTCF, we have performed a similar analysis of this data separating ATAC-seq peaks by CTCF-binding rather than YY1-binding. This shows a similar trend to YY1, but is overall less pronounced, and is now included in our analysis records. We have reported previously that loss of CTCF from many binding sites on Xi requires SmcHD1 (Gdula et al, 2019).

14. When the authors use cf. do they simply mean see also, or as wikipedia suggests: "the cited source supports a different claim (proposition) than the one just made, that it is worthwhile to compare the two claims and assess the difference". Perhaps it would be worth spelling out to clarify for the audience.

We used “cf.” in the text to mean “compare with”, when referring to a plot/observation/piece of data outside of the figure being immediately discussed (either in another study or different section of the paper). We were not aware of the recommendation to only use the cf abbreviation when the two items are intended to be contrasted. We do not believe this to be a universal grammatical convention, but nevertheless have changed incidences of cf. to “see also”.

Reviewer #1 (Significance (Required)):

General assessment: An important question in human biology is how much the sex chromosome contributes to sex differences in disease frequency. Genes that escape X inactivation in humans seem to have considerable impact on gene expression genome-wide. While there are not as many genes in mouse that escape inactivation, the use of the mESC cell differentiation approach allows detailed assessment of the timing of silencing during inactivation. The authors utilize an inter-specific cross and it would be interesting to know the limitations of such a system (in terms of informative DHS/genes that are informative).

Advance: As the authors note, there are multiple studies of similar systems that have revealed differences in the speeds of silencing of genes. However, this is the first study to my knowledge that has then tried to assess timing with gene-specific factors. There are multiple studies in humans comparing escape and subject genes for TFs, but lacking the developmental timing that this study incorporates.

Audience: While generally applicable to a basic research audience interested in gene regulation, the applicability to human genes that escape inactivation may interest cancer researchers or clinical

Full Revision

audiences interested in sex differences.

Reviewer #2 (Evidence, reproducibility and clarity (Required)):

The authors studied the molecular basis of a variation in the rate of individual gene silencing on the X undergoing inactivation. They took advantage of ATAC-seq to observe the kinetics of chromatin accessibility along the inactive X upon induction of Xist expression in mESCs. They demonstrated a clear correspondence between the decrease in chromatin accessibility and the silencing of nearby genes. Furthermore, they found that persistently accessible regulatory elements and slow-silencing were associated with binding of YY1. YY1 tended to associate longer with genes that required more time to be silenced than those that became silenced fast on the inactive X during XCI. The acute loss of YY1 facilitated silencing of slow genes in a shorter period. They suggest that whether or not the transcription factors stay associated longer is another factor that impacts the variation in the rate of gene silencing on the Xi.

Reviewer #2 (Significance (Required)):

It has been suggested that the rate of gene silencing during XCI varies depending on the distance of individual genes from the Xist locus or the entry site of Xist RNA on the X, as well as their initial expression levels before silencing. This study provides another perspective on this issue. The persistent association of transcription factors during XCI affects the rate of gene silencing. Although the issue addressed here might draw attention from only the limited fields of specialists, their finding advances our understanding of how the efficiency of silencing is controlled during the process of XCI. The experimental data essentially support their conclusion, and the manuscript was easy to follow. However, I still have some comments, which I would like the authors to consider before further consideration.

Major concerns

1. Based on the results shown in Figure 3E and F, the authors concluded that YY1 was more resistant than other TFs against the eviction from the X upon Xist induction. I am not still convinced with this. YY1 binds DNA via the zinc finger domain, while Oct4 binds DNA via the homeodomain. The difference in the binding module between them might affect their dissociation or the response to Xist RNA-mediated chromatin changes. In addition, given that YY1 has been reported to bind RNA, including Xist, as well, Oct4 might not be a good TF to compare.

We acknowledge and agree that our singular comparison between YY1 and OCT4 is insufficient to support a general conclusion that YY1 is unique with respect to its binding properties on Xi. This was also alluded to by Reviewer #1 (see 10.), where in response we write about the difficulties of selecting other appropriate/feasible candidate TFs for ChIP-seq in order to widen the comparison beyond OCT4. In consideration of this concern, we have re-phrased our conclusions regarding this point in the text, both at the point where it is first presented (Fig3F) and in the first discussion paragraph.

Full Revision

Furthermore, the difference in allelic ratio change between YY1 and OCT4 is admittedly not dramatic, and this metric can be influenced somewhat by the properties of the sets of peaks used (which is also why we have not tried to add statistical significance to this comparison in Fig3F). In order to make the comparison with OCT4 (a classic pluripotency factor), we were also limited to using mESC culture without differentiation conditions. It is possible that more pronounced differences between YY1 and other TFs would be observed under conditions where XCI is able to proceed further.

Even so, we contend that our observation that YY1 binding is lost from the Xi relatively slowly likely stands without a requirement for a comparison with OCT4 or other transcription factors. The decrease in allelic ratio for YY1 ChIP occurs more slowly than overall loss of chromatin accessibility from REs, which is arguably a more general proxy for TF binding, and much slower than kinetics of gene silencing (Fig3D and FigS2C). In addition, no other TF motifs (except CTCF, which has its own unique properties) were found significantly enriched within persistently-accessible REs, which would be an expectation if a different factor had similar properties of late-retained Xi binding as YY1.

Thus, overall we have tried to write the paper without overstating in isolation the importance of our claim that YY1 binding on Xi is relatively resistant to Xist-mediated inactivation, instead emphasising that it should be considered alongside the other pieces of data in the study.

2. I don't think that Kinetics of YY1 eviction upon Xist induction in SmcHD1 KO cells during NSC differentiation fit the phenotype of Smchd1 mutant cells. Although their previous study by Bowness et al (2022) showed that Smchd1-KO cells fail to establish complete silencing of SmcHD1-dependent genes, their silencing still reached rather appreciable levels according to Figure 6 of Bowness et al (2022). This is, in fact, consistent with the idea that XCI initially takes place in the mutant embryos, at least to an extent that does not compromise early postimplantation development. On the other hand, a significant portion of YY1 appears to remain associated with the target genes on both active and inactive X (Figure S4), which I think suggests that the presence of YY1 is compatible with silencing of SmdHD1-dependent genes. This is contradictory to the proposed role of YY1 that sustains the expression of X-linked genes in this context.

At any given timepoint of XCI, our data sets of gene silencing (ChrRNA-seq) consistently show a more pronounced allelic skew compared to chromatin accessibility (ATAC-seq). This behaviour is discussed in relation to Figure 1 in the text (see Results paragraph 2). We do not wish to overinterpret this quantitative difference because the assays are technically different and accessibility is not linearly correlated with gene expression. With this in consideration, we interpret the ATAC-seq data presented in Figure S4 to be fully consistent with the iXist-ChrX SmcHD1 KO ChrRNA-seq data in Figure 6 of our previous publication ie. a small increase in residual Xi gene expression from SmcHD1 KO NPCs is accompanied by a more appreciable increase in residual Xi chromatin accessibility. In line with this, it would not be contradictory for substantially increased Xi YY1 binding to sustain a quantitatively small (but nonetheless meaningful) increase in residual gene expression from Xi.

Full Revision

Figure S1C (this study)

Figure 6C (previous study)

Figure S1C (this study)

Additionally, the context in which we include this SmcHD1 KO ATAC-seq data in the current paper is to hypothesise a potential role for SmcHD1 in contributing towards the eventual removal of YY1 binding from Xi. This hypothesis is essentially based on two observations; 1.) There is substantially more residual YY1 binding to Xi in mESC no diff conditions (Figure 3) and 2.) One difference between no diff and diff conditions is absence of SmcHD1 recruitment in the former (Figure 5 in our previous study). The new SmcHD1 KO ATAC-seq data adds a third observation which supports the hypothesis - that YY1-bound REs are appreciably more accessible from Xi in SmcHD1 KO. However, none of these observations are direct evidence of a link between SmcHD1 and YY1, and more experiments would be required to substantiate this potential mechanism. If confirmed, it would be logically reasonable to suggest a role for YY1 in contributing towards the residual expression of X-linked in the context of SmcHD1 KO, but we do not yet claim this, and a potential link with SmcHD1 KO is not the main focus of the paper.

Reviewer #3 (Evidence, reproducibility and clarity (Required)):

In this manuscript, Bowness and colleagues describe the interesting finding that the transcription factor YY1 is associated with slow silencing genes in induced X-Chromosome Inactivation (XCI). The authors have conducted a comprehensive characterization of X-linked gene silencing and the loss of chromatin accessibility of regulatory elements in induced XCI in ESCs and during NPC differentiation. X-linked gene silencing was classified into four categories, ranging from fast-silenced genes to genes that escape silencing. Motif enrichment analysis of regulatory elements associated with slowly silenced genes identified YY1 as the transcription factor most significantly enriched. The separation of YY1-target and non-target genes confirmed that most genes bound by YY1 indeed exhibit slower silencing kinetics. A comparison of the binding kinetics of YY1 to another transcription factor, OCT4, during XCI revealed that YY1 is evicted more slowly compared to OCT4 on the inactive X, suggesting that slower eviction is a unique property of YY1. Conditional knock-outs of YY1 using protein degradation during induced XCI in mESCs demonstrated that the loss of YY1 at

Full Revision

target genes enhances silencing. This supports the hypothesis that YY1 serves as a crucial barrier for slow-silenced genes during XCI. Finally, the authors propose a hypothesis regarding the mechanism of YY1 eviction, suggesting a potential connection to the role of SmcHD1 during XCI.

The authors provide an in-depth analysis of the role of YY1 in gene silencing kinetics during induced XCI and believe this manuscript should be published if our comments are addressed.

Major comment:

Based on the allelic ratio in figure 3C only minor loss of YY1 binding occurs in induced XCI in mESCs on the Xi, while silencing is established properly as shown in figure 4C (left panel, red boxplots). This suggests that YY1 eviction is not necessarily required for these genes to be effectively silenced. Could the authors explain this discrepancy in the data regarding their manuscript conclusions? It seems this is true for XCI happening during differentiation towards NPCs, but not if cells are stuck in the pluripotency stage?

Whilst indeed substantial, we do not consider the silencing seen for 6-day mESCs in Fig4C to be “established properly”. We refer to our previous publication (Figure 4 Bowness et al., 2022), which shows that silencing at equivalent timepoints under differentiation conditions (d5-d7) is significantly more pronounced (near-“complete”). Indeed, the level of silencing reached by YY1-FKBP mESCs (Xist induced but no dTAG treatment) aligns with the plateau of silencing in undifferentiated mESCs we describe in our previous study (median allelic ratio of approximately 0.1).

We conclude that YY1 contributes somewhat to sustaining this residual expression in mESCs, because a) substantial YY1 binding remains on Xi at these timepoints in mESCs and b) silencing increases with degradation of YY1 (the latter is more direct evidence). Notably, silencing does not progress to completion (allelic ratio of 0) in the absence of YY1, so we do not claim that YY1 is the *only* factor sustaining residual Xi gene expression in mESCs.

We interpret this comment to be a fundamentally similar concern to that raised by Reviewer #2 (2.), but in the context of undifferentiated mESCs rather than SmcHD1 KO. As stated above, we do not think it inherently contradictory for substantially increased Xi YY1 binding to sustain a quantitatively small (but nonetheless meaningful) increase in residual gene expression from Xi.

Minor comments:

1. In the abstract lines 7-8, the authors state that the experiments were performed in mouse embryonic stem cell lines, but much of the data shown is acquired in NPC differentiations. Please adjust abstract.

We have adjusted this sentence in the abstract to include that many of the experiments in the paper involved differentiation of iXist-ChrX mESCs.

2. The last sentence of the abstract states that YY1 acts as a barrier to silencing but as stated in my major comment, that does seem to be the case in ESC differentiation towards NPCs, but not in

Full Revision

ESCs themselves. Please tone down this sentence. Moreover, we do not fully understand where the 'is removed only at late stages' comes from? Is this because of the Smchd1 link? We find this link quite weak with the data presented. We would tone down that last abstract sentence.

We have toned down the final sentence of the abstract accordingly. We agree that “removed *only* at late stages” is unsubstantiated since YY1 binding on Xi decreases over the entire time course (albeit slowly). However, we maintain that a connection between YY1 and late stages of the XCI process is reasonable to infer from the various pieces of evidence we provide in the study (egs YY1 is persistently enriched in accessible REs, it is associated with slow-silencing genes, and it remains bound to Xi in undifferentiated mESCs).

3. Several comparisons to human XCI have been made in the article. We do agree that there are similarities between mouse and human XCI. However, there is insufficient data that substantiates that these genes are regulated in a similar manner in humans. We believe the comparisons should be removed altogether or attenuated.

We agree that there is nothing in our data that directly pertains to human XCI. Comparisons to human are only made twice in the paper: Initially in the introduction to make a broad statement that many mechanisms of Xist function are conserved between species, and finally as speculation in the last discussion paragraph. We think it is relevant to acknowledge the parallels between our study, which links YY1 binding with resistance to Xist-silencing in a mouse ESC model, and literature describing a similar association between YY1 and XCI escape in humans.

4. At bottom of page 4, the authors say that for any given gene, the allelic ration of accessibility at its promoter decreased more slowly than it silenced and then write Fig 1B. They probably mean S1C? Since 1B only shows 4 genes.

The phrase “any given” was used colloquially (ie imprecisely), so we have replaced it with “individual”.

5. Figure 1B shows the average allelic ratio of multiple clones for genes representing different silencing speeds. Each data point is the average of multiple clones for these representative genes, could the authors show the individual data points or the standard deviation?

Fig1B predominantly shows the averages of only two replicate time-courses of Xist induction with NPC differentiation using the same parental clonal cell line, iXist-ChrX-Dom, but performed on different dates and passages. We regenerated the panel without merging the replicate data points, but this has little effect on the plot (see the Rmarkdown html file of Figure 1 on Github).

6. Figure 1B. Loss of promoter accessibility lags behind loss of chromatin-associated RNA expression for these 4 genes. What about distal REs? Do the allelic ratios for the distal REs more closely follow chromatin-associated RNA expression? Could the authors show this in a supplemental figure?

Full Revision

We comment from FigS1C on the general trend that accessibility decrease from Xi occurs slower than gene silencing (measured by ChrRNA-seq). We then find in FigS1D that distal elements lose accessibility slightly faster than promoters. Although overall the allelic ratio decrease of distal (non-CTCF) RE accessibility is slightly closer to the trajectory to that of gene silencing, it remains substantially slower (see again the Rmarkdown .html file of Figure 1 on Github).

An equivalent plot to Fig1B showing distal REs would rely on our simplistic assignment of distal elements to their nearest genes. We believe this is reasonable generalisation for investigating chromosome-wide trends but unlikely to be sufficiently accurate at the level of specific genes.

7. Figure 1B: gene silencing trajectory is depicted left while the legend says right. Same for promoter accessibility.

The legend is now corrected.

8. Figure S1A shows only part of the X chromosome. The area downstream of Xist is missing. Is this because the iXist-ChrX_{Dom} cell line is missing allelic resolution as shown in figure S2A? Could the authors explain in the figure legend that part of the X-Chromosome is missing?

We have now included a reference to the recombination event in the iXist-ChrX_{Dom} cell line both when we present data from this background in the first paragraph of the Results section, and in the legend of FigS1A.

9. Figure 2C shows that 94 TSSs bear a YY1 peak, yet Fig 2F shows 62 are targets of YY1. Is this because the rest are not properly silenced or are escapees?

Fig2C shows the numbers of ChrX YY1 ATAC-seq peaks which overlap with “promoters” (ie regions +/- 500bp of a TSS). By contrast, Fig2F shows ChrX genes classified as direct YY1-targets for allelic silencing analysis. The discrepancy between these numbers is due to a number of reasons:

- a) It is possible for multiple YY1 peaks to overlap the same promoter (eg one peak overlaps 500bp upstream, a separate peak overlaps 500bp downstream).
- b) The count in Fig2C is not restrictive to one TSS per gene in cases where there are multiple transcript isoforms in the gene annotation, thus multiple YY1 peaks can overlap different promoters for the same gene.
- c) A few genes do not pass our filters for allelic silencing analysis (eg they are too lowly expressed). Some YY1 peaks may overlap these genes.

We hope the revised version of Fig2F, which includes numbers of direct YY1 target genes on autosomes and ChrX, makes the distinction between these two numbers clearer.

10. Moreover, YY1 has ~4-fold more peaks on the X chromosome on distal elements compared to promoters. Yet figure 2F exclusively shows the proportion of YY1 binding sites on TSSs. Would distal REs show similar proportions for the silencing categories? Could the authors show the differences in a Supplemental figure?

Full Revision

As discussed in the response to Reviewer #1 (point 8.), a large fraction of distal YY1 peaks on ChrX are at LINE1 elements, which are not amenable to allelic analysis. Excluding these peaks results in a smaller number of distal elements bound by YY1. The application of our filters for allelic analysis reduces the number of distal YY1-bound REs even more, and our assignment of distal REs to their nearest gene is imprecise. For these reasons, we do not think a comparison of genes classified by whether they are putative targets of distal YY1-bound enhancers is informative.

11. The authors switch between different model systems in the figures, which makes quite confusing which type of XCI is being discussed. We would like to see clearly stated above all panels which cell culture condition is being studied (mESCs or NPCs).

We have tried to improve this potential source of confusion by modifying “mESC” to “mESC no diff” in the relevant figure panels (see response to Reviewer #1 comment 7B), and adding “in mESCs without differentiation” to the title of Figure 4.

12. In Figure 3E and 3F the authors look at the binding retention of OCT4 during XCI in ESCs. However, it is not clear why the authors choose OCT4. Could the authors explain why specifically OCT4 was chosen for these analyses?

In our responses to the other reviewers, we discuss the limitations of only having one other TF to compare to YY1. The choice of OCT4 was primarily dictated by our experience and confidence in being able to generate high quality ChIP-seq data of this factor.

As it was essentially arbitrary for the purposes of this paper, we have added a comment to this effect in the text (“with that of a different arbitrary TF, OCT4”).

13. What is the expression level of YY1 in NPCs compared to mESCs? In Supplemental S2A, it seems that YY1 protein levels decrease over time during NPC differentiation. Is part of the increased eviction a result of lower protein levels of YY1? Probably not since you calculate ratios between Xi and Xa. Can you please comment on this?

We were similarly intrigued by this apparent decrease in YY1 protein levels in NPCs (there is no decrease on the RNA level) and initially considered if it could contribute to the relative.

In FigS2A specifically, the d18 NPC band is probably just a poor quality sample extraction. Our ChIP-seq data generated from the same sample is similar poor compared to the others (FigS2B). In other YY1-FKBP12^{F36V} clones we derived and characterised by Western (not described further in this study, but will likely be published as raw source data for the cropped blots we show in FigS2A), the apparent difference in YY1 protein levels in NPCs is less pronounced. Although a minor decrease in YY1 protein in NPCs seems to be robust, we do not think it relevant in the context of our analysis of YY1 and XCI, as we almost always use Xa as internal comparison for any observations made about Xi.

14. On page 7 the authors state that degrading YY1 does not affect Xist spreading and/or localisation. Indeed, it has been previously shown by other groups that YY1 is required for Xist

Full Revision

localisation during XCI. Could the authors elaborate further on the why their cells behave differently compared to the Jeon 2011 paper?

We are working with a mouse ESC model of inducible Xist from its endogenous locus on ChrX and using the dTAG system to degrade YY1 protein. By contrast, Jeon 2011 worked with an Xist transgene integrated at random in the genome of mouse embryonic fibroblasts (MEFs) and siRNA knockdown of YY1. The difference in our observations could be linked to any of these 4 differences (ie cellular context, *Xist* genomic location, Xist introns, knockdown strategy), but we cannot identify a specific explanation.

15. In figure 4G and figure S3D elevated levels of Xist are observed in the dTAG conditions. As the authors point out, this could then result in accelerated silencing of the X seen upon YY1 loss. Are these elevated Xist levels that result in enhanced silencing in figure 4 relevant for the kinetics of silencing? Moreover, YY1 could act as transcriptional regulator of those genes in the X and by removing YY1, one would expect decreased transcription, which would be read as accelerated silencing. The authors could see whether the genes that show accelerated silencing are regulated by YY1 in ESCs (+ dTAG, - Dox).

We agree that these points are important to consider when interpreting the results of the YY1-FKBP12^{F36V} ChrRNA-seq we present in Figure 4. However, we believe they are covered in the text during our discussion of the data.

In relation to the final suggestion, the silencing of almost all X-linked genes is increased upon YY1 removal so separating a specific set of genes which show accelerated silencing would be difficult. Nevertheless, in Fig4F we report that the increases in Xi silencing are strongest for direct YY1 target genes. In fact, these genes also show a minor decrease in expression in the + dTAG - Dox condition (see response to Reviewer #1 point 12.). However, by-and-large the differences in Xa log2FCs between YY1-target and non-target genes are less statistically significant. Non-significant p-values are not shown on Fig4F, but can be found in our Rmarkdown analysis records.

16. Can the authors explain why they decided to put the Smchd1 part after the conclusion? Before the conclusion would have been better? The probable link between YY1 and SmcHD1 is definitely something important to investigate.

Supplemental FigS4 relating to SmcHD1 is more speculative and we lack direct mechanistic evidence linking YY1 and SmcHD1. It would require more experiments to substantiate this as a mechanism. We think these experiments could potentially be very interesting, but are beyond the scope of this study.

17. In the paper the authors cite Bowness et al., 2022. In it, Figure 5F studies silencing times with respect to silencing dependency on SmcHD1. What is the overlap between SmcHD1 target genes and YY1 target genes? This would provide more data about the correlation between YY1 and SmcHD1.

Full Revision

There is an association between YY1 target genes and our previous categories of genes based on SmcHD1 dependence (13/56 SmcHD1_dependent genes are YY1 targets compared to only 8/101 of SmcHD1_not_dependent genes). However, this enrichment of YY1 targets in SmcHD1 dependent genes is not so striking to warrant inclusion into the (very short) discussion of SmcHD1 in this paper. This association is also expected from the fact that both YY1-target genes and SmcHD1-dependent genes associate with the set of slow-silencing genes.

Of note, our categories of SmcHD1 dependency were in fact defined in a previous study (Gdula et al., 2019) from a different cellular model (SmcHD1 KO MEFs).

18. The authors hypothesise that SmcHD1 might play a role in the eviction of YY1 in NPC differentiation. The current data shows impaired silencing of slow silencing genes and YY1-dependent genes in the SmcHD1 knock-out. However, it doesn't show SmcHD1 is required for YY1 eviction. Could the authors provide direct evidence for their hypothesis by performing NPC differentiation in wild type and SmcHD1 knock-out cells and investigate YY1 binding using ChIP-seq?

The data we show in FigS4 is ATAC-seq data. It shows that YY1 target REs are particularly more accessible from the Xi in SmcHD1 KO, which is not direct evidence but does align with a potential role for SmcHD1 in mediating removal of YY1 binding from Xi (see our response to Reviewer #2's comment 2.). We agree that YY1 ChIP-seq over the same time course would be an interesting experiment, but arguably this would also only be indirect evidence (ie increased Xi YY1 enrichment may be due to a confounding consequence of SmcHD1 KO). We therefore believe the full suite of experiments needed to rigorously test the hypothesis are beyond the scope of this paper.

19. In figure S4A and S4B no significance is indicated among the different conditions across the different differentiation days. Could the authors add this?

At all timepoints, differences of Xi accessibility between YY1-binding vs non-YY1 REs are significant. P values are now added to FigS4 and the statistical test is described in the legend.

20. Finally, we would like the authors to elaborate in the conclusion about the order of events. As they correctly state at the top of page 5 (and we agree), delayed loss of promoter accessibility compared to gene silencing does not automatically mean that it is downstream of gene silencing. Can you elaborate on this? Also, in light of Fig S2C where loss of YY1 binding seems to happen after gene silencing.

We mention in the text and in the above response to Reviewer #2 (point 2.) that we do not wish to overinterpret this quantitative difference because the assays are technically different and accessibility is not linearly correlated with gene expression.

It is possible to speculate plausible biological explanations for this discrepancy in kinetics between accessibility loss, TF binding and gene silencing. For example, a change in the landscape of histone modifications at a promoter may have little effect on its accessibility to TFs but directly hinder RNA

Full Revision

Polymerase II in initiation and/or elongation of transcription of the gene. However, we prefer to keep this speculation out of the main text of the paper.

Reviewer #3 (Significance (Required)):

This manuscript highlights a novel role for YY1 in XCI. The manuscript provides an analysis of the correlation and causation of YY1 in gene silencing during XCI. There is a clear correlation between YY1 and delayed silencing of genes on the Xi. To our knowledge, this is the first time such an analysis has been performed for YY1. It advances our conceptual and mechanistic understanding of gene silencing kinetics and what the factors involved in it are. We believe it is an important contribution to the XCI field and will be of great value to the XCI community.

Strength:

This study presents a comprehensive and in-depth characterization of X-linked gene silencing during XCI.

Two different types of inducible XCI are studied and compared (ESCs vs differentiation towards NPCs), which we are grateful for.

Systematic and stepwise analysis of the data is very strong.

Many data points have been collected which provide stronger conclusions.

Weakness:

Some sentences in the abstract should be toned down.

YY1 eviction on the inactive X doesn't seem crucial to establish X-linked gene silencing in mESCs.

The mechanistic approach at the end of the manuscript with relation to SmcHD1 could be studied further.

This paper will be suited for a specialised audience in XCI and transcription factor control of gene expression, i.e. basic research.

Field of expertise: XCI, epigenetics, Xist, gene silencing, X chromosome biology.

Dear Neil,

Thank you for the submission of your revised manuscript. We have now received the enclosed reports from the referees that were asked to assess it, and I am happy to say that both support the publication of your study now.

Only some more minor editorial requests will need to be addressed before we can proceed with the official acceptance of your manuscript.

- Please also submit the final ms text as a word file without the figures. All main and EV figures need to be uploaded as high quality and individual figure files. You can find more info about our figure file types in our guide to authors online: <https://www.embopress.org/page/journal/14693178/authorguide#expandedview>
- The nomenclature and callouts for the Supplemental Figures need to be changed to Expanded View Figures (Figure EV1, Figure EV2, etc.).
- The Supplemental Tables S1-S5 are called out and their legends are in the ms, but the actual tables are missing. Please rename the tables depending on their type, e.g. Table 1, Table EV1, or Dataset EV1. You can find more info about our table files in our guide to authors: <https://www.embopress.org/page/journal/14693178/authorguide#expandedview>
- Please add up to 5 keywords to the ms file.
- Please correct the conflict of interest subheading to "Disclosure and Competing Interests Statement"
- Please correct the REFERENCE FORMAT: et al needs to be used after 10 author names. The EMBO reports reference style is also in EndNote.
- Please co-submit with the final ms a completed author checklist that can be found here: <https://www.embopress.org/page/journal/14693178/authorguide>
The completed checklist will also be part of our transparent peer-review process file.
- Please address these comments by our data editors:
 1. Please indicate the statistical test used for data analysis in the legend of supplementary figure 3b.
 2. Please note that the box plots need to be defined in terms of minima, maxima, centre, bounds of box and whiskers, and percentile in the legends of figures 1d; 2e, g-h; 3c, f; 4c-d, f, supplementary figures 1d; 2b; 4a-b.
 3. Please note that information related to n is missing in the legends of figures 1d; 2e, g-h; 4c-d, f, supplementary figure 1d.
 4. Although 'n' is provided, please describe the nature of entity for 'n' in the legends of supplementary figures 4a-b.

EMBO press papers are accompanied online by A) a short (1-2 sentences) summary of the findings and their significance, B) 2-3 bullet points highlighting key results and C) a synopsis image that is exactly 550 pixels wide and 200-600 pixels high (the height is variable). You can either show a model or key data in the synopsis image. Please note that text needs to be readable at the final size. Please send us this information along with the final manuscript.

Best,
Esther

Referee #1:

My comments have been addressed by the authors. I suggest this manuscript be published in EMBO reports and congratulate the authors for this very interesting paper!

Referee #2:

The responses to the issues I raised in the first round of review are good and I have no further comments. Regarding the issues raised by reviewer 1, I think that the authors have not only responded to his/her requests and modified the data presentation and some statements, but also provided reasonable explanations that would address the concerns raised. Regarding comment 2, I do not think that the authors should necessarily include a gene-by-gene comparison table as supplementary data.

All editorial and formatting issues were resolved by the authors.

Prof. Neil Brockdorff
University of Oxford
Biochemistry
New Biochemistry
South Parks Road
Oxford, Oxfordshire ox1 3qu
United Kingdom

Dear Neil,

I am very pleased to accept your manuscript for publication in the next available issue of EMBO reports. Thank you for your contribution to our journal.

Best,
Esther
